# WeatherArchive-Bench: Benchmarking Retrieval-Augmented Reasoning for Historical Weather Archives

## Abstract

Historical news segments on weather events are collections of enduring primary source records that offer rich, untapped narratives of how societies have experienced and responded to extreme weather events. These qualitative accounts provide insights into societal vulnerability and resilience that are largely absent from meteorological records, making them valuable for climate scientists to understand societal responses. However, their vast scale, noisy digitized quality, and archaic language make it difficult to transform them into structured knowledge for climate research. To address this challenge, we introduce WeatherArchive-Bench, the first benchmark for evaluating end-to-end retrieval-augmented generation (RAG) systems on historical weather archives. WeatherArchive-Bench comprises two tasks: *WeatherArchive-Retrieval*, which measures a system's ability to locate historically relevant news segments from over one million archival news segments, and *WeatherArchive-Assessment*, which evaluates whether Large Language Models (LLMs) can classify societal vulnerability and resilience indicators from extreme weather narratives and answer queries using the segments retrieved. Extensive experiments across sparse, dense, and re-ranking retrievers, as well as a diverse set of LLMs, reveal that dense retrievers often fail on historical terminology, while LLMs frequently misinterpret vulnerability and resilience concepts. These findings highlight key limitations in reasoning about complex societal indicators and provide insights for designing more robust climate-focused RAG systems from archival contexts. The constructed dataset and evaluation framework are publicly available at: https://anonymous.4open.science/r/WeatherArchive-Bench/.

## 1 Introduction

Extreme weather events are becoming increasingly frequent and severe as a result of climate change, posing urgent challenges for climate adaptation and disaster preparedness (O'Brien et al., 2006). Climate policymakers are expected to design targeted adaptation strategies that integrate disaster response with long-term planning, including climate-resilient urban development (Xu et al., 2024) and sustainable land use policies (Zuccaro et al., 2020). Achieving these goals requires not only meteorological data, but also a deeper understanding of how communities, infrastructures, and economic sectors have responded to climate hazards (Bollettino et al., 2020; Mallick et al., 2024). Historical archives provide such knowledge, documenting past extreme weather events alongside their cascading economic impacts, community responses, and local adaptation practices (Carey, 2012; Yu et al., 2025). A systematic analysis of these records can reveal which factors were most disruptive during a specified extreme weather event, thereby providing evidence-based insights to inform future climate policy interventions.

Recent advances in generative AI provide the feasibility to process substantial collections and extract structured insights at scale. In particular, RAG is a methodology that combines information retrieval systems with generative language models to enhance performance on knowledge-intensive domain tasks (Lewis et al., 2020). In terms of the climate domain, RAG systems can search through vast collections of weather records (Tan et al., 2024) and then interpret retrieved information into structured insights about climate impacts and societal responses (Vaghefi et al., 2023; Xie et al., 2025). RAG thus serves as an essential approach for climate adaptation planning, helping policymakers systematically learn from past weather impacts to enhance current risk assessment and decision-making (Brönnimann et al., 2019). However, applying RAG to historical climate archives for extracting societal vulnerability and resilience insights is non-trivial. For instance, historical documents contain outdated terminology, Optical Character Recognition (OCR) errors, and narrative formats that mix weather accounts with unrelated text (Bingham, 2010; Verhoef et al., 2015). Such noise and irregular structures pose challenges for both retrieval systems in locating relevant passages and LLMs in interpreting societal insights. Similarly, LLMs face significant knowledge gaps when processing historical archives, since these data are generally unavailable online and excluded from pre-training corpora (Liu et al.,

Table 1: Comparison of existing QA benchmarks with WEATHERARCHIVE-BENCH.

| Dataset | # Papers | Paper Source | Domain | Historical Data | Task |
|---------|----------|--------------|--------|-----------------|------|
| REPLIQA | 17.9K | Synthetic | General | ✗ | Topic Retrieval + QA |
| CPIQA | 4.55K | Climate papers | Climate Sci. | ✗ | Multimodal QA |
| ClimRetrieve | 30 | Reports | Climate Sci. | ✗ | Document Retrieval |
| ClimaQA | 23 | Textbooks | Climate Sci. | ✗ | Scientific QA |
| WeatherArchive | 1.04M | Hist. archives | Climate Sci. | ✓ | Retrieval + QA + classification |

2024a). Thus, models might not be able to interpret historical terminology for vulnerability and resilience analysis. Likewise, existing retriever models might also struggle to identify climate-relevant passages, especially when historical vocabulary and narrative styles differ substantially from modern web content (Perełkiewicz & Poświata, 2024). More importantly, no existing benchmark systematically evaluates RAG performance on historical climate archives, which constrains the possibility of optimizing the RAG systems in the climate domain. As shown in Table 1, existing benchmarks focus on relatively small-scale paper source and primarily target scientific papers and reports rather than historically grounded archival data. None evaluates the extraction of societal vulnerability and resilience indicators for climate adaptation planning. This evaluation gap prevents the development of AI systems capable of translating irreplaceable historical evidence of past climate responses into actionable intelligence for contemporary policy decisions.

To address this gap, we introduce WEATHERARCHIVE-BENCH, the first large-scale benchmark for retrieval-augmented reasoning on historical weather archives. WEATHERARCHIVE-BENCH focuses on two complementary tasks: *WeatherArchive-Retrieval*, which evaluates retrieval models' ability to identify evidence-based passages in response to specified extreme weather events, and *WeatherArchive-Assessment*, which measures LLMs' capacity to answer evidence-based climate queries and classify indicators of societal vulnerability and resilience from archival narratives using the retrieved passages. In this context, vulnerability refers to the susceptibility of communities, infrastructures, or economic systems to climate-related harm, while resilience denotes the capacity to absorb and recover from climate shocks (Feldmeyer et al., 2019). Understanding these dimensions from historical records is critical for identifying risk factors (Rathoure, 2025), designing interventions, and learning from past adaptation strategies across contexts and time periods (Kelman et al., 2016). To support rigorous evaluation, we curate over one million OCR-parsed archival documents with dedicated preprocessing strategies, followed by expert validation and systematic quality control. We then evaluate a range of retrieval models and state-of-the-art LLMs on three core capabilities required for climate applications: (1) processing archaic language and noisy OCR text typical of historical documents, (2) understanding domain-specific terminology and concepts, and (3) performing structured reasoning about socio-environmental relationships embedded in narratives. Our results reveal significant limitations of current systems: dense retrieval models often fail to capture historical terminology compared to sparse methods, while LLMs frequently misinterpret vulnerability and resilience indicators. These findings highlight the need for methods that adapt to historical archival data, integrate structured domain knowledge, and reason robustly under noisy conditions.

In summary, our contributions are threefold:

1. We introduce WEATHERARCHIVE-BENCH, which provides two evaluation tasks: *WeatherArchive-Retrieval*, assessing retrieval models' ability to extract relevant historical passages, and *WeatherArchive-Assessment*, evaluating LLMs' capacity to classify societal vulnerability and resilience indicators from archival weather narratives.

2. We release the first large-scale corpus of over one million historical archives, enriched through preprocessing and human curation to ensure quality, enabling both climate scientists and the broader community to leverage historical data.

3. We conduct comprehensive empirical analyses of state-of-the-art retrieval models and LLMs on historical climate archives, evaluating them within a fully end-to-end RAG pipeline. This exposes key limitations in handling archaic language and domain-specific terminology and provides concrete insights for building more robust, retrieval-grounded climate QA systems.

## 2 RELATED WORK

### 2.1 CLIMATE-FOCUSED NLP

The urgency of addressing environmental challenges has intensified in recent decades, driven by mounting evidence of climate change, habitat degradation, and biodiversity loss (Rathoure, 2025). Advancing disaster preparedness requires tools that can assess vulnerabilities and resilience using realistic, context-rich cases, which urban planners and policymakers can directly act upon (Birkmann et al., 2015; Jetz et al., 2012). Recent climate-focused NLP systems demonstrate impressive technical capabilities. ClimaX (Nguyen et al., 2023) achieves competitive performance with operational forecasting systems by pre-training on diverse CMIP6 meteorological datasets, while ClimateGPT (Thulke et al., 2024) integrates interdisciplinary climate research to outperform general-purpose models on climate-specific benchmarks. Specialized models, such as WildfireGPT (Xie et al., 2024), employ RAG for hazard-specific tasks, like wildfire risk assessment. Yet, these advances focus largely on contemporary climate data and physical processes. They overlook real-world impact records (e.g., detailed accounts of infrastructure failures, agricultural losses) embedded in archival sources. Such records are essential for climate policy makers and urban planners to understand long-term vulnerability patterns and inform policies aimed at future disaster preparedness.

### 2.2 BENCHMARKING IN CLIMATE AI

Historically, progress in climate AI has been constrained by the scarcity of large-scale, practical datasets that capture real-world climate impacts in sufficient temporal and geographic breadth (Zha et al., 2025). Existing resources predominantly target either physical climate modelling tasks or narrowly scoped contemporary text analysis, leaving historical, case-based impact records largely untapped. For example, ClimateIE (Pan et al., 2025) offers 500 annotated climate science publications mapped to the GCMD+ taxonomy, yet focuses on technical entities such as observational variables rather than societal consequences of extreme weather. Evaluation benchmarks face similar limitations. ClimateBench (Watson-Parris et al., 2022) and those catalogued in ClimateEval (Kurfalı et al., 2025) primarily assess meteorological prediction accuracy or general climate communication tasks, with little insight into how well language models can identify and interpret vulnerability and resilience patterns from complex, domain-specific archives. As Xie et al. (2025) shows that RAG offers a promising avenue for climate experts to locate relevant cases within vast collections to support analysis in the context of extreme events, yet its effectiveness in systematically capturing historical vulnerability and resilience information remains untested. This gap in both data and evaluation motivates our introduction of WEATHERARCHIVE-BENCH, a large-scale benchmark for assessing AI systems' capacity to extract and synthesize real-world climate impact narratives.

## 3 WEATHERARCHIVE-BENCH

Our goal with WEATHERARCHIVE-BENCH is to provide a realistic benchmark for evaluating current retrieval and reasoning capabilities in the context of climate- and weather-related archival texts. In particular, we focus on the dual challenges of (i) constructing a high-quality corpus from archival news segments and (ii) defining retrieval and generation tasks that capture the practical needs of climate researchers. This section details our corpus collection pipeline and task formulation.

### 3.1 CORPUS COLLECTION

LLMs are generally pre-trained on large-scale internet corpora, which frequently include fake and unreliable content (Roy & Chahar, 2021). In contrast, archival news segments provide a unique and valuable information source, as copyright restrictions typically exclude them from LLM pretraining data. Unlike standardized meteorological datasets, archival news segments provide rich, narrative descriptions of weather-related disruptions and community-level adaptation successes (Sieber et al., 2024). These archives also capture public voices and societal perspectives that would be prohibitively expensive to collect today, yet public perceptions remain documented in historical records (Thurstan et al., 2016). Thus, our corpus offers contextualized insights that complement traditional climate data. For climate scientists seeking to understand long-term patterns of societal vulnerability and resilience, these narrative-rich sources provide invaluable evidence of how communities have historically experienced, interpreted, and responded to weather-related challenges.

Our corpus construction emphasizes both scale and reliability. Sourced from a proprietary archive institution, we collected two 20-year tranches of news archives from an organization in Southern Quebec, a region representative of broader Northeastern American weather patterns: one covering a contemporary period (1995–2014) and one covering

a historical period (1880–1899). The archival news articles were digitized with OCR and subsequently cleaned using GPT-4o, following the post-OCR correction method of Zhang et al. (2024). Although OCR noise is a known issue in archival processing, retaining it would distort our evaluation of climate comprehension and cross-document reasoning, which are the core goals of our benchmark. Unlike OCR-focused datasets such as OHRBench Liu et al. (2024b) that vary noise levels to study error cascades, our benchmark intentionally provides OCR-corrected text to isolate climate-specific retrieval and societal-impact reasoning. The detailed preprocessing and validation steps are provided in Appendix A. We then segmented the archival news articles into overlapping archival news segments using a sliding-window approach, followed by the method proposed by (Sun et al., 2023), allowing each segment to preserve sufficient semantic context while satisfying token-length constraints. The resulting dataset comprises 1,035,862 news segments, each standardized to approximately 256 tokens, which we used for the *WeatherArchive-Retrieval* task creation.

## 3.2 TASK DEFINITION

WEATHERARCHIVE-BENCH incorporates two complementary tasks designed to mirror the workflow of climate scientists. *WeatherArchive-Retrieval* tests models' ability to locate relevant historical evidence. The other is *WeatherArchive-Assessment*, which evaluates their capacity to interpret complex socio-environmental relationships within an archival report of an extreme weather event.

### 3.2.1 WEATHERARCHIVE-RETRIEVAL

Figure 1: The construction pipeline of the retrieval task in weather archive collections. The process integrates newspaper collection, keyword frequency search, and human verification to construct a high-quality corpus of weather-related archival news articles with relevance judgments for each query.

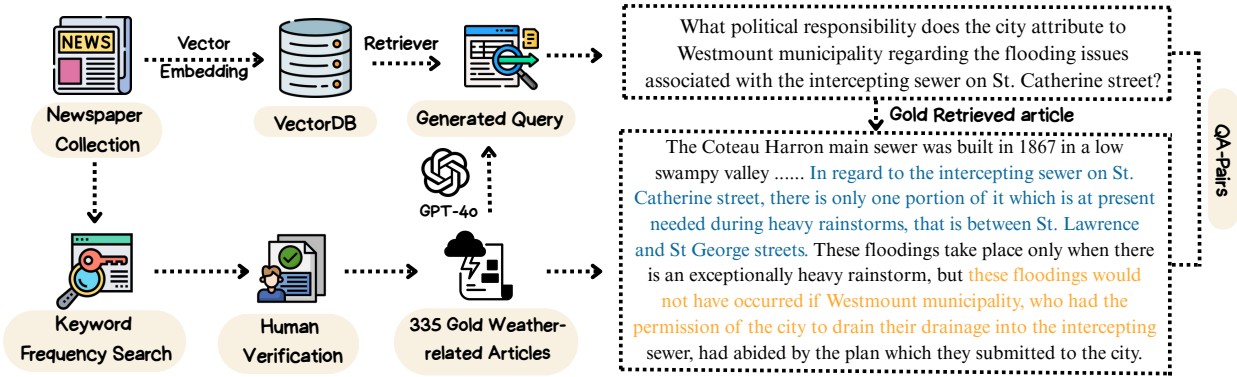

In scientific domains such as climate analysis, scientists often rely on precedents embedded in long historical archives (Herrera et al., 2003; Slonosky & Sieber, 2020; Sieber et al., 2022). A well-designed retrieval task (Figure 1) is essential, as it evaluates a model's ability to identify contextually relevant and temporally appropriate information while providing a reliable foundation for subsequent answer generation.

To construct the benchmark, we first ranked 1,035k archival news segments by the frequency of keywords related to disruptive weather events, as detailed in Appendix B.1. From this ranking, we selected the top 525 segments, which were then manually reviewed by domain experts to identify those providing complete evidential support for end-to-end question answering. After curation, 335 high-quality validated segments were retained. To better characterize their topical coverage, we analyzed the word frequencies of these segments and present the results as a word cloud in Appendix B.2. For each passage, we generated domain-specific queries using GPT-4o, with prompts provided in Appendix C. These queries were designed to emulate real-world research intents, resulting in a realistic retrieval benchmark composed of query–answer pairs.

The difficulty of this task stems from the nature of the segments extracted from historical archives. Unlike contemporary datasets, news archives use domain-specific terminology that has shifted over time (e.g., outdated expressions for storms or floods; presented in Appendix D), which makes relevance judgments nontrivial. Moreover, archival news articles frequently embed descriptions of weather impacts within broader narratives or unrelated sections such as advertisements, which introduces additional noise into the retrieval process. By grounding evaluation in such historically situated and noisy data, *WeatherArchive-Retrieval* establishes a challenging yet realistic testbed for assessing the robustness of retrieval models and systems.

### 3.2.2 WEATHERARCHIVE-ASSESSMENT

To effectively support climate scientists in disaster preparedness, language models must go beyond retrieving relevant news segments and demonstrate the ability to interpret societal vulnerability and resilience as documented in historical texts. To this end, we design an evaluation framework to assess a model's ability to reason about climate impacts across multiple levels, drawing on established approaches from vulnerability and adaptation research (Feldmeyer et al., 2019). An overview of the societal vulnerability and resilience framework is provided in Appendix E. The framework comprises two complementary subtasks: (i) classification of societal vulnerability and resilience indicators, and (ii) open-ended question answering to assess model generalization on climate impact analysis. To clarify the task setup, we summarize the dataset construction here: historical newspaper narratives are paired with daily weather records from the same location and period. Each example includes a climate-related query and retrieved passages reflecting the documented weather impact. Archival news articles are digitized, cleaned, and aligned with meteorological data so both sources describe the same event. Questions require models to link the narrative with the corresponding weather record, and answers are evaluated against these aligned labels. Our experimental pipeline follows a retrieve-then-answer setup. The retriever selects the most related archival news segments using the query exactly as written, since all archival texts are already chunked. The QA model then generates its answer solely from these retrieved segments. There are 335 examples in the WeatherArchive-Assessment task, with a one-to-one mapping between each historical weather report and its corresponding query, resulting in 335 query–answer pairs. The construction pipeline is illustrated in Figure 2. A more detailed description of the human validation process is provided in Appendix F, and ground-truth oracles are generated using GPT-4.1 with structured prompting, as detailed in Appendix G.

Figure 2: *WeatherArchive-Assessment* - the construction pipeline of assessment task on societal vulnerability and resilience. GPT-4.1 evaluates retrieved archival news segements across multiple criteria, with human verification ensuring quality before generating ground truth answers. This sample case shows the assessment of rainstorm impacts.

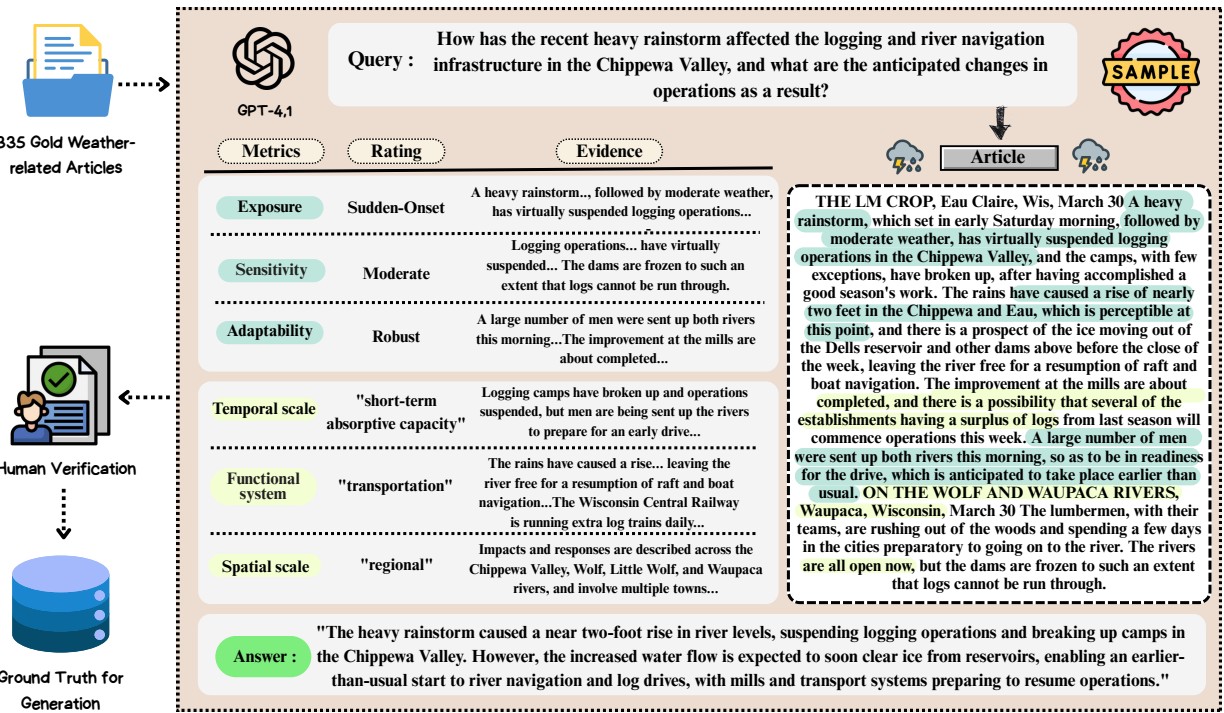

**Societal Vulnerability.** Vulnerability is widely conceptualized as a function of exposure, sensitivity, and adaptability (O'Brien et al., 2004). We operationalize this framework by prompting models to assign descriptive levels to each component. Prompt details are provided in Appendix G.2. Specifically, *exposure* characterizes the type of climate or weather hazard, distinguishing between sudden-onset shocks (e.g., storms, floods), slow-onset stresses (e.g., prolonged droughts, sea-level rise), and compound events involving multiple interacting hazards. *Sensitivity* evaluates how strongly the system is affected by such hazards, ranging from critical dependence on vulnerable resources to relative insulation from disruption. *Adaptability* captures the ability of the system to respond and recover, spanning robust governance and infrastructure to fragile conditions with little or no coping capacity.

This classification-wise evaluation examines whether models can move beyond surface-level text interpretation toward structured reasoning about vulnerability, which is essential for anticipating climate risks (Linnenluecke et al., 2012). In practice, *exposure* and *adaptability* are often signalled by explicit indicators (Brooks et al., 2005) such as infrastructure damage or recovery measures, which evaluate LLMs' capacity to capture through climate factual extraction. *Sensitivity* is more challenging, as it requires climate reasoning (Montoya-Rincon et al., 2023) about governance quality, institutional strength, or social capital, factors that are seldom directly expressed in segments. By incorporating both explicit and implicit aspects of vulnerability, our framework provides a rigorous test of whether models can integrate factual evidence with contextual inference.

**Societal Resilience.** Resilience is evaluated using indicators proposed by Feldmeyer et al. (2019), which emphasize adaptation processes across three scales. On the *temporal scale*, models must distinguish between short-term absorptive capacity (e.g., emergency response), medium-term adaptability (e.g., policy or infrastructure adjustments), and long-term transformative capacity (e.g., systemic redesign). On the *functional system scale*, models identify which systems are affected, including health, energy, food, water, transportation, and information, highlighting their interdependence in shaping preparedness. Lastly, on the *spatial scale*, models assess resilience across levels (e.g., local, community, regional, national), capturing variation in adaptability across contexts. Through the experts' annotation process, we are informed that temporal indicators are often easier to identify since newspapers tend to report immediate damages and responses explicitly, whereas functional and spatial dimensions are more challenging since they require models to infer systemic interactions and contextual variation that are rarely stated explicitly in news archives. By formulating these criteria into multiple-choice questions, we evaluate whether models can recognize structured indicators of resilience within noisy archival narratives.

## 4 EXPERIMENTAL SETUP

### 4.1 EVALUATION METRICS

**WeatherArchive-Retrieval.** We evaluate retrieval performance with the commonly used metrics, including Recall@$k$, MRR@$k$, and nDCG@$k$ for $k \in \{3, 10, 50, 100\}$. These metrics capture complementary aspects of performance: coverage (Recall), top-rank relevance (MRR), and graded relevance (nDCG).

**WeatherArchive-Assessment.** The downstream benchmark evaluates model performance on climate-related reasoning tasks via expert-validated reference standards. Evaluation proceeds along two dimensions: (i) Vulnerability and resilience indicator classification: models must identify and categorize societal factors from historical weather narratives, with performance quantified through F1, precision, and recall metrics to measure classification accuracy; (ii) Historical climate question answering: models generate responses to evidence-based climate queries using retrieved news segments, with answer quality assessed via BLEU, ROUGE, and BERTScore for semantic similarity to expert-authored responses, supplemented by token-level F1 between predictions and ground truth. Additionally, we employ LLM-based judgment using GPT-4.1 to evaluate climate reasoning quality beyond traditional similarity metrics, determining whether model-generated responses contain factual errors. Outputs are compared against oracle answers and are judged correct if the model's response to the specified climate-related question matches or encompasses the oracle answer and incorrect otherwise. These metrics jointly test whether models can both recognize structured indicators and reason accurately in free-form responses, aligning evaluation with the benchmark's goal of measuring climate-relevant retrieval-augmented reasoning.

**Question Answering (QA) in Climate AI.** Each climate-related query is first passed through the retrieval component, which returns the top-3 news segments that serve as the model's only reference material. This setup allows us to evaluate end-to-end RAG performance, where generation quality depends directly on retrieval quality. This design captures real-world RAG scenarios where retrieval quality directly impacts generation quality. When retrieved segments lack relevant information, models must explicitly acknowledge this limitation rather than hallucinate answers. Gold-standard answers are generated separately for evaluation purposes only, ensuring we can measure both retrieval effectiveness and the LLM's ability to work with imperfect retrieval results. While vulnerability and resilience classification measures the ability to extract structured evidence, free-form QA tests whether models can synthesize dispersed archival information and articulate climate impacts in support of scientific reasoning.

### 4.2 EVALUATED MODELS

**Retrieval Models.** We evaluate a set of retrieval models on the archival collections, including three categories: (i) sparse lexical models: BM25 (BM25plus, BM25okapi) (Robertson et al., 2009) and SPLADE (Formal et al., 2021)

Figure 3: Examples of ground-truth in WEATHERARHIVE-ASSESSMENT for a historical snowstorm event

```
Question – What specific financial loss did the Toronto Telephone Company
incur due to the snowstorm on February 21, and how did the storm affect their
infrastructure?
-------------------------------------------------------------------------------

Archival news segments – Trains were late today in consequence of the heavy
snowstorm last night.  Toronto, February 21.  The snowstorm early this morning
did great damage.  The wires of the Toronto Telephone Company were blown down
from the JO ail building.  Their losses alone will amount to $1,000.  Thorold,
Ont, February 21.  A storm of rain and sleet set in at midnight last night,
covering the ground about three inches thick.  Travel not impeded.  Peterborough,
Ont, February 21.  Snow fell last night and this morning to the depth of four
inches.  A snowstorm set in at three o'clock this afternoon.  Sleighing good;
travel unimpeded.  Brampton, Ont, February 21.  A wild storm set in last night.
The roads in the country are said to never have been worse.  Chatham, Ont,
February 21.  Weather very stormy today.  Heavy northwest wind, with snow.  Roads
muddy and almost impassable.  Grimsby, February 21.  The most violent storm of
the season is now prevailing here

-------------------------------------------------------------------------------

Classification Ground-Truth

region:Toronto, exposure:Sudden-Onset, sensitivity:Critical
adaptability:Constrained, temporal:short-term absorptive capacity,
functional:information, spatial:local

-------------------------------------------------------------------------------

Question-Answering Ground-Truth

Answer:  The Toronto Telephone Company incurred a financial loss of $1,000
due to the snowstorm on February 21.  The storm caused significant damage to
their infrastructure, specifically by blowing down their wires from the JO ail
building.
```

(ii) dense embedding models: ANCE (Xiong et al., 2020), SBERT (Reimers & Gurevych, 2019), and large propri-etary embeddings, including OpenAI's text-embedding-ada-002 (Neelakantan et al., 2022), Gemini's text-embedding (Lee et al., 2025), IBM's Granite Embedding (Awasthy et al., 2025), and Snowflake's Arctic-Embed (Yu et al., 2024) and (iii) re-ranking models: cross-encoders applied on BM25 candidates (BM25plus+CE, BM25okapi+CE) with a MiniLM-based reranker (Wang et al., 2020; Hofstätter et al., 2020). Implementation details for each model are pro-vided in Appendix J.

**Language Models.**   We consider a diverse suite of open-source and proprietary LLMs with various parameter scales. Open-source models include Qwen-2.5 (7B–72B), Qwen-3 (4B, 30B), LLaMA-3 (8B, 70B), Mistral-8B and Ministral-8×7B. These families capture scaling effects, efficiency–performance trade-offs, and robustness to long or noisy text. We also include DeepSeek-V3-671B, which targets efficient scaling and adaptability. Proprietary models include GPT (3.5-turbo, 4o), Claude (opus-4-1, sonnet-4) and Gemini-2.5-pro, which are widely used in applied pipelines, offering strong reasoning and summarization capabilities. All models are instruction-tuned versions, denoted as "IT". The computational costs of deploying these models are provided in Appendix H.

## 5   EXPERIMENTAL RESULTS

In this section, we evaluate and analyze the performance of retrieval models and state-of-the-art LLMs on *WeatherArchive-Retrieval* and *WeatherArchive-Assessment* tasks in our benchmarks.

## 5.1 WEATHERARCHIVE-RETRIEVAL EVALUATION

Table 2: Retrieval performance (in percentages) on *WeatherArchive-Retrieval* across sparse, dense, and re-ranking models. **Bold** and underline indicate the best and the second-best performance.

| Category | Model | Recall (%) | | | | nDCG (%) | | | |
|---|---|---|---|---|---|---|---|---|---|
| | | @3 | @10 | @50 | @100 | @3 | @10 | @50 | @100 |
| **Sparse** | BM25PLUS | 58.5 | 67.8 | 79.1 | 82.7 | 49.7 | 53.2 | 55.7 | 56.3 |
| | BM25OKAPI | 54.3 | 67.8 | 79.4 | 83.0 | 44.4 | 49.4 | 51.9 | 52.5 |
| | SPLADE | 7.5 | 18.2 | 47.8 | 64.5 | 6.0 | 9.7 | 16.0 | 18.8 |
| **Dense** | SBERT | 29.0 | 40.0 | 50.1 | 55.2 | 22.8 | 26.8 | 29.0 | 29.8 |
| | ANCE | 34.0 | 52.2 | 77.9 | 86.6 | 27.3 | 33.8 | 39.4 | 40.8 |
| | ARCTIC | 53.4 | 67.5 | 82.1 | 91.0 | 44.3 | 49.4 | 52.7 | 54.2 |
| | GRANITE | 54.6 | 71.9 | 88.7 | 94.6 | 44.8 | 51.2 | 55.0 | 55.9 |
| | OPENAI-3-SMALL | 51.6 | 67.8 | 85.4 | 91.9 | 43.6 | 49.4 | 53.3 | 54.4 |
| | OPENAI-3-LARGE | 48.1 | 65.1 | 85.4 | 92.2 | 40.0 | 46.1 | 50.7 | 51.8 |
| | OPENAI-ADA-002 | 51.0 | 70.2 | 88.1 | 95.5 | 42.1 | 49.2 | 53.1 | 54.3 |
| | GEMINI-EMBEDDING-001 | 57.3 | 74.9 | **91.6** | **95.8** | 47.9 | 54.3 | 58.2 | 58.8 |
| **Re-Ranking** | BM25PLUS+CE | **63.9** | **76.1** | 81.2 | 82.7 | 53.2 | 57.9 | 59.0 | 59.3 |
| | BM25OKAPI+CE | **63.9** | **76.1** | 81.8 | 83.0 | **53.3** | **58.0** | **59.2** | **59.4** |

**Sparse Retrieval Models Achieve Strong Top-rank Relevance on Climate Archives.** As shown in Table 2, BM25 variants continue to perform strongly, often matching or surpassing dense alternatives in ranking quality at top $k$. The effectiveness of BM25 might be related to the nature of climate-related queries, which usually contain technical terminology and domain-specific collocations (e.g., "flood damage," "hurricane casualties," "crop failure due to drought"). In such cases, exact lexical matching is critical as sparse methods are able to capture these specialized terms directly, whereas dense representations may blur over distinctions or concepts. For instance, a query about "storm surge fatalities" would benefit from precise overlap with news segments containing the same terminology, whereas a dense retriever might incorrectly emphasize semantically related but distinct expressions such as "storm warnings" or "storm intensity", as in an example case provided in Appendix C. These findings highlight the importance of sparse methods in scientific and technical domains where specialized vocabulary governs relevance.

**Re-ranking Procedure Could Deliver Better Performance.** With the effective sparse methods, further deploying a re-ranker could achieve better performance. In this setup, BM25 provides high lexical coverage at the candidate generation stage, and the re-ranker ranks the top candidates by modelling fine-grained query–document interactions. Empirically, the results show that hybrid models such as BM25plus+CE and BM25okapi+CE consistently outperform both pure sparse and pure dense baselines within the top-ranked results (e.g., top 3-10 segments), which are most critical for downstream QA. This indicates that re-ranking models with a baseline yield more robust performance for climate-related retrieval.

## 5.2 WEATHERARCHIVE-ASSESSMENT EVALUATION

**Factual Extraction vs. Climate Reasoning in Societal Vulnerability Assessment.** Consistent with prior work on scaling laws (Kaplan et al., 2020), larger models generally improve zero-shot generation. This is likely because greater capacity increases the chance of encountering relevant patterns during pretraining. Table 3 shows that Claude-Opus-4-1 achieves the best overall performance, while among open-source models, DeepSeek-V3-671B ranks highest, followed by Qwen2.5-72B-IT. Models perform well on explicit indicators of exposure and adaptability, such as infrastructure damage or recovery measures, where factual extraction is sufficient. On the other hand, sensitivity indicator classification relies more heavily on reasoning about how strongly a system is affected by weather stressors (Morss et al., 2011), including governance quality and social capital dimensions that are rarely explicit in archival news segments. While some proprietary models maintain strong performance on these tasks, open-source models show a sharper decline. Overall, larger models improve factual extraction and can handle certain reasoning tasks effectively, but challenges remain for tasks that require inferring implicit relationships with society and extreme weather from archival news segments.

**LLMs Struggle with Socio-environmental System Effects.** Societal resilience indicator classifications require recognizing direct damages from disruptive weather events and reasoning about how shocks propagate across geographic

Table 3: F1 evaluation results (in percentages) for vulnerability and resilience indicator classification on *WeatherArchive-Assessment* across diverse LLMs. **Bold** and underline indicate the best and second-best results.

| Model | Vulnerability (%) | | | Resilience (%) | | | Average (%) |
|---|---|---|---|---|---|---|---|
| | Exposure | Sensitivity | Adaptability | Temporal | Functional | Spatial | |
| GPT-4O | 64.6 | 52.8 | 58.0 | 62.3 | **64.5** | 51.8 | 59.0 |
| GPT-3.5-TURBO | 63.6 | 46.6 | 46.5 | 64.3 | 34.2 | 39.5 | 49.1 |
| CLAUDE-OPUS-4-1 | 78.3 | 67.6 | 67.5 | **84.6** | 62.5 | **61.4** | **70.3** |
| CLAUDE-SONNET-4 | 77.2 | **73.8** | 59.7 | 65.2 | 63.5 | 60.3 | 66.6 |
| GEMINI-2.5-PRO | 76.6 | 62.0 | 57.1 | 75.6 | 62.5 | 61.3 | 65.8 |
| DEEPSEEK-V3-671B | **79.8** | 49.5 | **70.9** | 76.0 | 61.3 | 60.8 | 66.4 |
| MIXTRAL-8X7B-IT | 27.3 | 21.4 | 24.1 | 32.2 | 21.4 | 32.6 | 26.5 |
| MINISTRAL-8B-IT | 43.7 | 18.8 | 24.6 | 45.8 | 41.9 | 37.0 | 35.3 |
| QWEN3-30B-IT | 65.8 | 44.4 | 30.0 | 73.0 | 34.2 | 36.4 | 47.8 |
| QWEN3-4B-IT | 32.0 | 27.5 | 18.4 | 49.6 | **64.5** | 28.5 | 36.8 |
| QWEN2.5-72B-IT | 74.4 | 43.4 | 67.6 | 73.5 | 49.8 | 51.5 | 60.0 |
| QWEN2.5-32B-IT | 53.3 | 31.2 | 44.9 | 60.9 | 46.9 | 36.5 | 45.6 |
| QWEN2.5-14B-IT | 40.5 | 39.2 | 29.5 | 35.7 | 23.4 | 30.3 | 33.1 |
| QWEN2.5-7B-IT | 33.8 | 9.1 | 22.5 | 33.0 | 30.8 | 32.9 | 27.0 |
| LLAMA-3.3-70B-IT | 36.7 | 42.9 | 24.4 | 48.1 | 53.1 | 35.5 | 40.1 |
| LLAMA-3-8B-IT | 24.3 | 19.8 | 18.4 | 19.4 | 29.0 | 28.6 | 23.3 |
| Average | 54.7 | 40.6 | 41.5 | 56.2 | 46.5 | 42.8 | 47.0 |

scales and interdependent systems. As shown in Table 3, models achieve relatively strong performance on temporal dimensions with a score of 0.562 on average, with Claude-Opus-4-1 and DeepSeek-V3-671B reliably identifying immediate response capacities. However, performance degrades on functional and spatial dimensions, where even sophisticated models struggle to assess cross-system dependencies (e.g., over-predicting "transportation" or "information") and multi-scale coordination (e.g., overlooking "local"). Samples are provided in Appendix L.2. Impacts are distributed unevenly across systems and exhibit inherently scale-dependent propagation dynamics. This pattern reveals limitations as models perform well at identifying direct impacts, yet are limited in reasoning over complex socio-environmental interdependencies that mediate systemic resilience. This highlights that multi-scale vulnerability assessment still requires human expertise.

Figure 4: Performance comparison of LLMs on free-form QA task across various metrics.

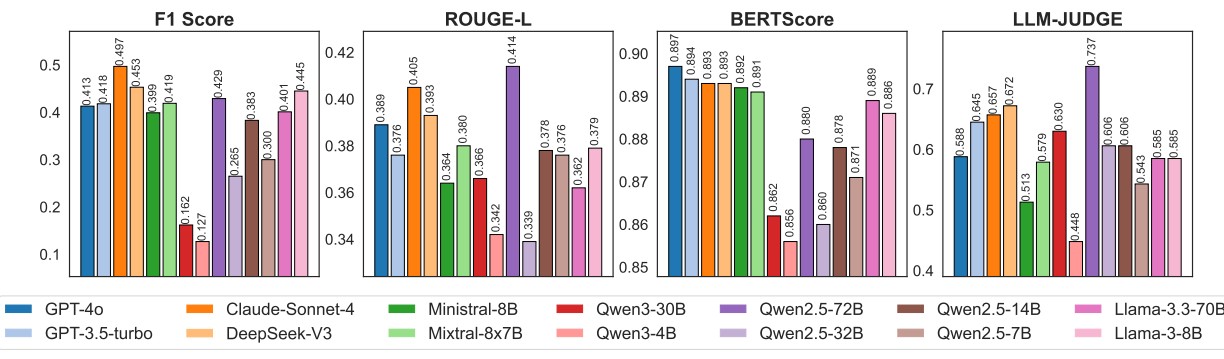

**From Retrieval to Reasoning: LLM Performance on Climate-specific QA.** To evaluate how retrieval-augmented LLMs translate historical climate records into actionable insights, we examine their ability to synthesize retrieved news segments into coherent, domain-specific answers. As shown in Figure 4, performance varies across metrics. Claude-Sonnet-4 achieves the highest lexical fidelity (BLEU: 0.141, F1: 0.497), while Qwen2.5-72B leads in semantic similarity (ROUGE-L: 0.737), reflecting stronger conceptual understanding despite lexical divergence. DeepSeek-V3 balances both aspects, offering robust factual grounding and semantic coherence. Token-level metrics emphasize the reliability of models with precise lexical alignment, whereas semantic evaluations reveal that open-source frontier models are narrowing the gap with proprietary systems by capturing nuanced reasoning needed for climate-specific queries. Overall, these results demonstrate that larger models can effectively integrate retrieved segments, yet generating scientifically accurate answers remains challenging, especially in the context of free-form climate-specific QA, with open-source systems still scaling toward both lexical fidelity and contextual reasoning. We further analyze the

impact of retrieval quality on these generation metrics in Appendix K.1, showing that better retrieval leads to higher question-answering performance.

## 6 CONCLUSION

WEATHERARCHIVE-BENCH establishes the first large-scale benchmark for evaluating the full RAG pipeline on historical weather archives. By releasing a dataset of over one million archival news segments, it enables climate scientists and the broader community to leverage historical data at scale. With well-defined downstream tasks and evaluation protocols, the benchmark rigorously tests both retrieval models and LLMs. In doing so, it transforms underutilized archival narratives into a standardized resource for advancing climate-focused AI.

Our analyses reveal that hybrid retrieval approaches outperform dense methods on historical vocabulary, while even proprietary LLMs remain limited in reasoning about vulnerabilities and socio-environmental dynamics. Future research should address two identified challenges: (1) enhancing retrieval methods to better handle historical vocabulary and narrative structures, and (2) improving models' ability to reason about complex socio-environmental systems beyond surface-level factual extraction. By offering a standardized evaluation resource, WEATHERARCHIVE-BENCH lays the groundwork for future research toward AI systems that can translate historical climate experience into actionable intelligence for adaptation and disaster preparedness.

## ETHICS STATEMENT

The WEATHERARCHIVE-BENCH is built from a collection of digitized historical newspapers provided through collaboration with an official organization, which remains anonymous at this stage. This organization retains the copyright of the archival news articles, but has granted permission to publish the curated benchmark in support of the climate research community. We additionally acknowledge that using GPT-based post-OCR correction may introduce model-driven biases, which we treat as an important consideration for the integrity of the task.

Although the majority of extreme weather events in our dataset are recorded in North America, the accounts capture how societies experienced and responded to climate hazards. These records provide broadly relevant insights into resilience strategies and adaptation planning that extend beyond their original geographical context. In addition, contributions from crowd-sourcing may be influenced by geodemographic factors, which introduces variation but also enriches the dataset (Hendrycks et al., 2020). As such, the benchmark reflects diverse societal perspectives on climate impacts and responses, making it a valuable resource for studying adaptation strategies across societal contexts.

## REPRODUCIBILITY STATEMENT

The WEATHERARCHIVE-BENCH dataset is publicly available and fully reproducible. While we cannot release the original newspaper print versions due to copyright restrictions, the post-OCR documents used in our benchmark are included in the supplementary material. Our complete codebase, including the data preprocessing pipeline and model evaluation scripts, is accessible through an anonymous GitHub repository. Reproducing experiments that involve proprietary components requires API keys for external services.

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

APPENDIX

## A  OCR CORRECTION QUALITY VALIDATION

Given the substantial noise inherent in OCR-digitized historical newspaper text, post-OCR correction was implemented as a critical preprocessing step to ensure corpus quality suitable for downstream applications. We employed GPT-4o with customized prompts to systematically correct OCR errors that could significantly impact text comprehension and retrieval performance.

### A.1  CORRECTION PROCESS

The post-OCR correction targeted several categories of common OCR artifacts:

- Character recognition errors (e.g., "COntInuInQ" → "continuing").
- Removal of extraneous artifacts and formatting noise.
- Standardization of spacing and punctuation.

---

**Before:**
OXTAEIO Ta os3cmo, Febrnarr 13, A collision oc-enrrel oc-enrrel on the Caada Simthern K iilwar, outit a mile and half wet of TUsonborg, t three 0 'dock this moruing, oetwaell twj t7 freight traios, coaaisting about 40 ar sach, mostlf ladett with ZG i-CraJ i-CraJ nercaaoxiuie, a4 low Wi ot pork, grain, tc. The men jumloed for their lives and av 0 ided the impending crash, which was terrific, both engnes being raised on their ends, and some cals telescoped and others crushed together inan inextricable mas s.

**After:**
OXTAEIO, Taos, February 13 A collision occurred on the Canada Southern Railway, about a mile and a half west of Tusonborg, at three o'clock this morning, between two freight trains, consisting of about 40 cars each, mostly loaded with ZG i-CraJ merchandise, and low with pork, grain, etc. The men jumped for their lives and avoided the impending crash, which was terrific, both engines being raised on their ends, and some cars telescoped and others crushed together in an inextricable mass.

---

### A.2  POST-OCR QUALITY ASSESSMENT

To validate the effectiveness of our automated correction pipeline, we conducted a systematic evaluation comparing GPT-4o's corrections against expert human annotations on a randomly selected sample of 50 articles from the corpus.

| Metric | n-gram | Score |
|---|---|---|
| BLEU | 1-gram | 0.911 |
| | 2-gram | 0.853 |
| | 3-gram | 0.817 |
| ROUGE | 1-gram | 0.947 |
| | 2-gram | 0.919 |
| | L | 0.943 |

Table 4: OCR correction quality comparing GPT-4o output against human annotations (n=50).

The consistently high scores across both BLEU and ROUGE metrics demonstrate strong alignment between automated and human corrections, validating the reliability of our preprocessing approach. These results confirm that the GPT-4o correction pipeline successfully preserves semantic content while eliminating OCR artifacts that would otherwise compromise retrieval accuracy and downstream task performance.

### A.3  IMPACT OF OCR NOISE ON RETRIEVAL PERFORMANCE

To address concerns about OCR preprocessing, we conducted additional experiments on the raw, uncorrected corpus to quantify the impact of OCR noise on retrieval performance.

Table 5: Retrieval performance comparison between raw OCR and corrected corpus. Results show substantial degradation under OCR noise, motivating our preprocessing pipeline.

| Retriever | nDCG@3 | nDCG@10 | nDCG@50 | nDCG@100 |
|---|---|---|---|---|
| BM25Plus | 0.259 | 0.315 | 0.376 | 0.392 |
| BM25Okapi | **0.265** | **0.323** | **0.385** | **0.424** |
| Arctic-Embed | 0.204 | 0.251 | 0.286 | 0.295 |
| ANCE | 0.104 | 0.139 | 0.175 | 0.185 |
| SBERT | 0.086 | 0.110 | 0.122 | 0.131 |
| SPLADE | 0.007 | 0.019 | 0.034 | 0.039 |

Compared with Table 2, OCR noise causes an average 47% reduction in nDCG@10 across all retrievers, with dense models experiencing greater degradation than sparse methods. These results confirm that uncorrected OCR artifacts would substantially compromise our ability to evaluate climate-specific retrieval and reasoning capabilities.

## B  HISTORICAL WEATHER ARCHIVE INFORMATION

### B.1  KEYWORDS IN WEATHER ARCHIVE

Our keyword taxonomy was developed through consultation with climate historians and concluded four primary categories that capture the multifaceted nature of weather-related archival content:

- **Natural Disaster Terms:** Direct references to weather-related hazardous events
- **Climate Phenomena:** Meteorological conditions and atmospheric processes
- **Geographic Context:** Spatial references that contextualize weather impacts
- **Societal Response:** Human and institutional responses to weather disruption

The expanded keyword list is presented in Table 6. These terms were finally counted based on their frequency in our historical archives to be selected for *WeatherArchive-Retrieval* task creation.

| Natural Disaster | Climate | Geographic Related | Support |
|---|---|---|---|
| Natural disaster | Extreme weather | Mountain area | Emergency response |
| Earthquake | Heavy rain | Coastal region | Evacuation |
| Flood | Snowstorm | River basin | Rescue operation |
| Hurricane | Hail | Urban flooding | Government aid |
| Tornado | Drought | Rural area | Disaster relief |
| Storm | Heat wave | Forest region | Support troops |
| Tsunami | Cold wave | | Civil protection |
| Landslide | Weather damage | | Humanitarian assistance |
| Wildfire | | | |
| Volcanic eruption | | | |

Table 6: Keywords used for frequency-based ranking of weather-related passages in historical archives.

### B.2  KEYWORDS WORDCLOUD

Figure 5 presents a word cloud visualization of the 335 curated weather archive passages. We used NLTK library to remove all punctuation and stop-words, and generated using TF-IDF weighting to visualize the distinctive terms characterizing weather-related disruptions and societal responses.

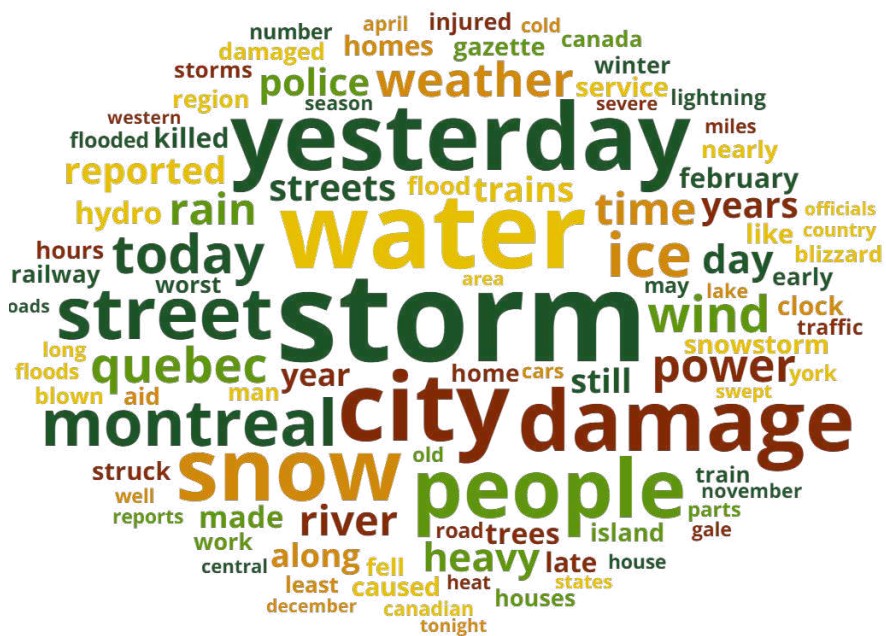

Figure 5: Word cloud for keywords in weather archives

This visualization reveals several prominent semantic clusters: direct meteorological phenomena ("storm," "rain," "snow," "water"), infrastructure and impact terms ("damage," "city," "power," "road"), temporal markers ("yesterday," "time," "year"), and societal elements ("people," "police," "reports"). Notably, the prominence of terms like "yesterday" and other temporal references reflects the immediacy of news reporting, while location-specific terms such as "montreal," "quebec," and geographic descriptors ("river," "island") emphasize the regional focus of the Southern Quebec archives. The prevalence of both historical ("railway," "telegraph") and modern infrastructure terminology validates our dataset's temporal span.

## C    QUERY GENERATION - *WeatherArchive-Retrieval*

We employed GPT-4o to generate climate-specific queries for each of the 335 curated gold passages. The prompt template ensures query specificity and uniqueness by requiring that each question can only be answered using information from its corresponding passage, and the example demonstrates how this approach creates challenging retrieval scenarios that require precise matching of quantitative details and geographic references within noisy archival text.

---

**Query Generation - Prompt**

Given the following passage about {Weather Type}, generate a single, focused question that meets these criteria:
1. Can be answered using ONLY the information in this passage
2. Focuses on the weather impact based on the given location
3. Is detailed and specific to this exact situation
4. Requires understanding the passage's unique context
5. Cannot be answered by other similar passages about {Weather Type}

Passage:
{Actual Passage}

---

---

**Query Generation - Example Result**

**What was the economic impact of the July 1996 flooding in the Saguenay, Quebec, on the insurance industry, and how did one company's commercial-property claims contribute to this situation?**

The most expensive winter storm recorded in Canada from an insurance perspective was a March 1991 tornado that tore through Sarnia, Ont., and caused $25 million worth of insurable damage. None of Montreal's previous ice storms made the insurance bureau's list of the most costly Canadian natural disasters. "Two people who work with me had tree branches crash through their roofs," Medza said the heaviest storm damage occurs during the summer months, when high winds can cause tornadoes and excessive rain brings sewer backups and floods homes. Of the 48 most costly Canadian storms, 31 of them occurred during the months of July and August. But it was on Sept. 7, 1991, that a severe hail storm rained down on the city of Calgary, causing $342 million in insurable damage to homes and cars - the highest total recorded by any natural disaster in Canada's history. Quebec's worst insurance bill was for $212 million after the July 1996 flooding in the Saguenay - although about $108 million of that total came from one company reporting three commercial-property claims. The cost of disaster Most expensive storms Cities Claims Amount paid Calgary, Alta. (hail) Sept.

---

## D  DIACHRONIC SHIFTS IN WORD USAGE

| Word | Definition | WeatherArchive samples | current use | Note |
|------|-----------|------------------------|-------------|------|
| **inundation** | "Great flood, overflow." Collocations: *inundation committee*. | The scheme was dropped as of no use, and the chairman of the **inundation** committee got home again. NOW THE RIVER. Saturday night's advices from down river ports were that at Three Rivers the lake ice was on the move since 2 p.m. | **flood**, **flooding** | Modern English prefers flood; *inundation* is now rare outside technical reports. |
| **tempest** | "Violent storm," with literary flavor. | A temporary panic was caused among the audience at the Port St. Martin Theatre by the sudden quenching of the gas light. Cries were raised of "Turn off the gas." The slamming of doors by the wind and the roar of the **tempest** drowned the voices of the actors. | **storm**, **severe storm**, **hurricane**, **cyclone** | Neutral, precise words dominate in reporting; *tempest* is archaic/poetic. |

Table 7: Comparison of historical words *inundation/tempest* and their modern equivalents.

## E  FACTORS INFLUENCING SOCIETAL VULNERABILITY AND RESILIENCE

The vulnerability and resilience frameworks presented in Tables 8 and 9 provide theoretically grounded and empirically validated dimensions for assessing climate impacts, as demonstrated in prior studies (Kelman et al., 2016). These

frameworks are particularly well-suited for evaluating LLMs because they translate abstract concepts into concrete, observable indicators that can be systematically identified in historical texts. Moreover, their hierarchical structure enables fine-grained assessment of model capabilities, ranging from surface-level factual extraction (e.g., identifying damaged infrastructure) to more complex reasoning about systemic interactions (e.g., understanding cross-scale governance coordination). The frameworks' established validity in climate research ensures that WEATHERARCHIVE-BENCH evaluates genuine climate reasoning capabilities rather than arbitrary categorization tasks.

Table 8: Factors influencing climate vulnerability. (Source: (Smit et al., 2000; McCarthy, 2001; Feldmeyer et al., 2019))

| | |
|---|---|
| **Exposure** | The degree of climate stress upon a particular unit of analysis. Climate stress can refer to long-term changes in climate conditions or to changes in climate variability and the magnitude and frequency of extreme events. |
| **Sensitivity** | The degree to which a system will respond, either positively or negatively, to a change in climate. Climate sensitivity can be considered a precondition for vulnerability: the more sensitive an exposure unit is to climate change, the greater are the potential impacts, and hence the more vulnerable. |
| **Adaptability** | The capacity of a system to adjust in response to actual or expected climate stimuli, their effects, or impacts. The latest IPCC report (McCarthy et al., 2001, p. 8) identifies adaptability as "a function of wealth, technology, education, information, skills, infrastructure, access to resources, and stability and management capabilities". |

Table 9: Factors influencing climate and disaster resilience. (Source: (Laurien et al., 2022; Keating et al., 2017; Mochizuki et al., 2018))

| | |
|---|---|
| **Temporal Scale** | Resilience capacity across time horizons: *short-term absorptive capacity* (immediate emergency response and coping mechanisms), *medium-term adaptability* (incremental adjustments and learning processes), and *long-term transformative capacity* (fundamental system restructuring and innovation for sustained resilience under changing conditions). |
| **Functional Scale** | Resilience across critical infrastructure and service systems: *health* (medical services and public health systems), *energy* (power generation and distribution), *food* (agricultural systems and food security), *water* (supply and sanitation systems), *transportation* (mobility and logistics networks), and *information* (communication and data systems). |
| **Spatial Scale** | Resilience across governance and geographic levels: *local* (community and municipal capacity), *regional* (provincial and multi-municipal coordination), and *national* (federal policies and cross-regional resource allocation and coordination mechanisms). |

## F   WEATHERARCHIVE-ASSESSMENT ORACLE GENERATION AND VALIDATION

The *WeatherArchive-Assessment* task requires reliable ground-truth oracles to evaluate model performance on classifying societal vulnerability and resilience indicators. To construct these oracles, we employ GPT-4.1 with carefully designed structured prompting, enabling consistent extraction of categorical labels from archival narratives.

To validate the reliability of GPT-4.1–generated oracles, we recruited four independent domain experts. Each annotator was assigned a disjoint subset of 60 rows, ensuring diverse coverage of the dataset. The resulting annotations yield an overall inter-annotator agreement of $\kappa_{\text{Fleiss}} = 0.67$, which is generally interpreted as substantial agreement in the literature. Disagreements among annotators were subsequently resolved through expert adjudication to establish high-quality reference labels.

$$\text{Accuracy}(f) = \frac{1}{n} \sum_{i=1}^{n} \mathbf{1}\{f(x_i) = y_i\} = 0.82$$

$$\omega = \frac{1}{m} \sum_{j=1}^{m} \mathbf{1}\{H_{0j} \text{ is rejected}\}, \quad \text{where } \omega = 0.75 > 0.5$$

We further assess the alignment between GPT-4.1 and the expert annotations. GPT-4.1 achieves an accuracy of $0.82$ when compared against the adjudicated ground truth, highlighting its effectiveness in streamlining the annotation process. To strengthen this validation, we adopt the statistical framework proposed by Calderon et al. (2025), with parameters $\epsilon = 0.05$ and $\alpha = 0.05$, to rigorously compare GPT-4.1 outputs against human judgments. The evaluation shows a winning rate of $\omega = 0.75$, meaning GPT-4.1 outperforms the majority of human annotators in 75% of pairwise comparisons. Since $\omega > 0.5$, this result shows that GPT-4.1 achieves higher agreement with the reference annotations than the average individual annotator, thereby validating its suitability for generating oracles used in benchmark evaluation. This framing emphasizes alignment with the annotation protocol rather than any normative comparison to human expertise.

# G PROMPT DESIGN - *WeatherAchive-Assessment*

## G.1 ORACLE ANSWER GENERATION

### G.1.1 PROMPTS

---

**Oracle Answer - Prompt**

You are a climate vulnerability and resilience expert. Implement a comprehensive assessment following the IPCC vulnerability framework and multi-scale resilience analysis.

VULNERABILITY FRAMEWORK:
- **Exposure**: Characterize the type of climate or weather hazard.
• Sudden-Onset → Rapid shocks such as storms, floods, cyclones, or flash droughts
• Slow-Onset → Gradual stresses such as sea-level rise, prolonged droughts, or heatwaves
• Compound → Multiple interacting hazards (e.g., hurricane + flooding + infrastructure failure)

- **Sensitivity**: Evaluate how strongly the system is affected by the hazard.
• Critical → Highly dependent on vulnerable resources; likely severe disruption
• Moderate → Some dependence, but buffers exist; disruption noticeable but not catastrophic
• Low → Minimal dependence on hazard-affected resources; relatively insulated

- **Adaptability**: Determine the system's capacity to respond and recover.
• Robust → Strong governance, infrastructure, technology, and social capital; effective recovery likely
• Constrained → Some coping mechanisms exist but are limited, uneven, or short-lived
• Fragile → Very limited or no capacity to cope; likely overwhelmed without external aid or systemic transformation

RESILIENCE FRAMEWORK:
- **Temporal Scale**: Choose the primary focus among [short-term absorptive capacity (emergency responses) — medium-term adaptability (policy/infrastructure adjustments) — long-term transformative capacity (systemic redesign/migration)]
- **Functional System Scale**: Classify the single most affected system based on evidence. Options: [health, energy, food, water, transportation, information]. Consider redundancy, robustness, recovery time, and interdependence.
- **Spatial Scale**: Choose the primary level among [local — regional — national]. Highlight capacity differences across scales.

INSTRUCTIONS:
- Always classify using the provided categories only, citing evidence from the document chunk.
- Ensure all classifications and selections are supported by evidence.

INPUT:
Query: query
Retrieved Document Chunk: context

OUTPUT FORMAT (follow this exact structure):
Region: [Extract/infer geographic region]
Exposure: [Sudden-Onset — Slow-Onset — Compound]
Sensitivity: [Critical — Moderate — Low]
Adaptability: [Robust — Constrained — Fragile]
Temporal: [short-term absorptive capacity — medium-term adaptability — long-term transformative capacity]
Functional: [health — energy — food — water — transportation — information]
Spatial: [local — regional — national]

EXAMPLE OUTPUT:
Region: Montreal
Exposure: Slow-Onset
Sensitivity: Moderate
Adaptability: Robust
Temporal: medium-term adaptability
Functional: energy
Spatial: regional

Only output in the exact format above, using the exact categories as instructed. Do not include any additional text.

---

To verified the reliability of the LLM-generated labels, even if LLM judges have some degree of variability, it would be overly costly for us to create numerous independent assessments for each instance. To address this, we take added measures: LLM-generated oracle QA outputs are manually examined for quality and consistency. We also take 50 QA

instances, about 1/7 of the dataset, and make them available for four independent human annotators to review. This added step confirmed the oracle answers used for our assessment task are reliable.

### G.1.2 EXAMPLE

---

**Oracle Answer - Example**

query: What specific infrastructure and agricultural impact did the British steamer Canopus experience due to the heavy gales in the United Kingdom?"

passage: "STORMY WEATHER Heavy gales over the United Kingdom Bourne weather on the Atlantic Disastrous loss of cattle shipments London, February 18 The weather continues very unsettled over the whole of the United Kingdom, and gales are reported at several stations The heavy gale which has raged at Penzance for the past two days has somewhat abated The wind is now blowing strongly from the southwest and the barometer marks 28.70 inches The gale is still blowing at Liverpool, but it has moderated a little London, February 18 The British steamer Canopus, Captain Horsfall, which arrived at Liverpool yesterday from Boston, lost her boats and 247 head of cattle, and sustained other damages in consequence of heavy weather Sports and Pastimes Curling Stuarton, X8, February 18 The curling match between the Truro and Stuarton clubs, which took place here today, resulted in a victory for Stuarton, which places the club in the van as good curlers Quebec, February 18 The Quebec Curling Club Challenge Cup was played for at the rink, St Charles street today, by the Montreal Caledonia Curling Club and the Quebec Curling Club The play was excellent on both sides, Quebec winning by 18 shots FEDERALIST London, February 19 At 2 a.m. the following was the score in the six days' walking match: Brown 328, Hazael 280, and ""Limping"" Day 258, and going splendidly AQUATIC."

region: United Kingdom
exposure: Sudden-Onset
sensitivity: Critical
adaptability: Constrained
temporal: short-term absorptive capacity
functional: food
spatial: national

---

### G.2 LLM GENERATION GENERATION

The table below presents the prompt designed to let LLMs generate the insights of a disaster record with some detailed instructions on how it should incorporate context into the response.

### G.2.1 PROMPTS

---

**LLM Generation - Prompt**

You are an expert in climate vulnerability and resilience. Use the retrieved document chunks to provide a comprehensive answer to the query.

INSTRUCTIONS:
- Consider all retrieved document chunks together before answering.
- Synthesize the information; do not repeat text verbatim unless quoting is necessary for evidence.
- Ensure that your answer is **directly relevant to the query** and grounded in the provided context.
- Avoid adding information not supported by the provided chunks.
- If the retrieved chunks are not relevant to the query, should claim there is no mention in the provided contexts.

INPUT:
Query: query
Retrieved Document Chunks:
context

OUTPUT:
[Provide a concise, evidence-based answer to the query, integrating information from all chunks]

---

G.2.2 EXAMPLE

> **LLM Generation - GPT-4O Example Result**
>
> query:What specific infrastructure and agricultural impact did the British steamer Canopus experience due to the heavy gales in the United Kingdom?"
>
> "STORMY WEATHER Heavy gales over the United Kingdom Bourne weather on the Atlantic Disastrous loss of cattle shipments London, February 18 The weather continues very unsettled over the whole of the United Kingdom, and gales are reported at several stations The heavy gale which has raged at Penzance for the past two days has somewhat abated The wind is now blowing strongly from the southwest and the barometer marks 28.70 inches The gale is still blowing at Liverpool, but it has moderated a little London, February 18 The British steamer Canopus, Captain Horsfall, which arrived at Liverpool yesterday from Boston, lost her boats and 247 head of cattle, and sustained other damages in consequence of heavy weather Sports and Pastimes Curling Stuarton, X8, February 18 The curling match between the Truro and Stuarton clubs, which took place here today, resulted in a victory for Stuarton, which places the club in the van as good curlers Quebec, February 18 The Quebec Curling Club Challenge Cup was played for at the rink, St Charles street today, by the Montreal Caledonia Curling Club and the Quebec Curling Club The play was excellent on both sides, Quebec winning by 18 shots FEDERALIST London, February 19 At 2 a.m. the following was the score in the six days' walking match: Brown 328, Hazael 280, and ""Limping"" Day 258, and going splendidly AQUATIC.
>
> region:United Kingdom
> exposure:Sudden-Onset
> sensitivity:Critical
> adaptability:Constrained
> temporal:short-term absorptive capacity
> functional:transportation
> spatial:regional

G.3 ANSWER COVERAGE EVALUATION

The below table presents the prompt designed to compare standard answers with model-generated answers to ensure that the model output covers the core information of the standard answer.

G.3.1 PROMPTS

> **Coverage Evaluation Instructions**
>
> You are an expert evaluator. Compare the oracle answer with the generated answer and determine if the generated answer COVERS the key information stated in the oracle answer.
>
> Oracle Answer: oracle_answer
> Generated Answer: generated_answer
>
> Task: Determine if the generated answer COVERS the key information from the oracle answer.
>
> Consider:
> - Does the generated answer contain the main points from the oracle answer?
> - Is the information accurate and relevant?
> - Does it address the same question/topic?
>
> Output ONLY: "true" if it covers, "false" if it doesn't cover.

H COMPUTATIONAL COSTS

For large proprietary models, a one-time evaluation on our *WeatherArchive-Assessment* benchmark costs approximately $1.2 for GPT-4o, $2.0 for Gemini-2-Pro, and $3.6 for Claude-Sonnet. For open-source models, evaluations

were conducted on a system equipped with four NVIDIA RTX 4090 GPUs (156GB for each GPU). The relatively modest computational requirements highlight the accessibility of our benchmark to researchers with limited resources, while still supporting comprehensive evaluation of state-of-the-art models.

## I  ANNOTATION PROCESS

We shuffled randomly selected 60 passages from this sample pool of 300 passages for manual annotation by our four annotators. After annotation, we calculated the Kappa coefficient between annotation results to assess consistency. Simultaneously, we generated an overall annotation file by comparing the annotations from all four annotators. For cases with high consensus among annotators, the response from the annotator with the highest frequency was adopted as the final answer. Where significant discrepancies existed, expert adjudication was used.

Finally, we compared this consolidated annotation with the responses generated by GPT-4.1 and found an exceptionally high degree of alignment. Consequently, we conclude that the GPT-generated results are highly credible and trustworthy.

## J  RETRIEVAL MODELS SELECTED

1. **Sparse retrieval** We include BM25 (BM25plus, BM25okapi) (Robertson et al., 2009) and SPLADE (Formal et al., 2021). BM25 is a standard bag-of-words baseline that scores documents using term frequency and inverse document frequency, implemented as a high-dimensional sparse vector dot product. Following BEIR (Thakur et al., 2021), we use the Anserini toolkit (Lin et al., 2016) with default Lucene parameters ($k_1 = 0.9$, $b = 0.4$). SPLADE extends sparse retrieval by predicting vocabulary-wide importance weights from masked language model logits and applying contextualized expansion with sparse regularization (Bai et al., 2020), to capture semantically related terms beyond exact token overlap.

2. **Dense retrieval** We consider ANCE (Xiong et al., 2020), SBERT (Reimers & Gurevych, 2019), and large proprietary embeddings such as OpenAI's text-embedding-ada-002 (Neelakantan et al., 2022), Gemini's text-embedding (Lee et al., 2025), IBM's Granite Embedding model (Awasthy et al., 2025) and Snowflake's Arctic-Embed (Yu et al., 2024). These models encode queries and passages into dense vectors and score relevance via inner product. We adopt publicly available checkpoints for ANCE, SBERT, Arctic, while OpenAI embeddings are queried via API. The inclusion of large proprietary models reflects the increasing role of commercial LLM-derived embeddings in applied domain-specific retrieval pipelines.

3. **Re-ranking model** We evaluate reranking models using cross-encoders (BM25plus+CE, BM25okapi+CE) (Wang et al., 2020). Specifically, two BM25 models from Anserini first retrieve the top 100 documents, after which a cross-attentional reranker jointly encodes the query–document pairs to refine the ranking. Following Thakur et al. (2021), we employ a 6-layer, 384-dimensional MiniLM (Wang et al., 2020) reranker model for our retrieval task. The overall setup follows Hofstätter et al. (2020).

## K  ADDITIONAL RESULTS

### K.1  IMPACT OF RETRIEVAL QUALITY ON GENERATION.

Table 10 demonstrates the end-to-end nature of our RAG evaluation. Using GPT-4o as a representative model, we observe substantial performance degradation when moving from gold passages to retrieved passages, confirming that our benchmark captures the propagated effect of retrieval quality on generation. The low performance without context validates that the task requires retrieval-augmented reasoning rather than relying on parametric knowledge alone.

Table 10: End-to-end RAG evaluation showing how retrieval quality impacts generation performance

| Setting | BLEU | ROUGE-1 | ROUGE-L | BERTScore | F1 | LLM Judge |
|---|---|---|---|---|---|---|
| No Context | 0.018 | 0.185 | 0.168 | 0.846 | 0.120 | 0.033 |
| Top-3 Retrieved | 0.112 | 0.309 | 0.276 | 0.878 | 0.292 | 0.137 |
| Top-3 + Gold | 0.121 | 0.414 | 0.380 | 0.885 | 0.380 | 0.755 |
| Gold Only | **0.158** | **0.466** | **0.425** | **0.896** | **0.458** | **0.788** |

## K.2 RETRIEVAL EVALUATION

Table 11: Additional retrieval performance on *WeatherArchive-Retrieval* across sparse, dense, and re-ranking models. MRR@k results (k=1,3,5,10,50,100).

| Category | Model | MRR@1 | MRR@3 | MRR@5 | MRR@10 | MRR@50 | MRR@100 |
|---|---|---|---|---|---|---|---|
| **Sparse** | BM25PLUS | 0.3701 | 0.4662 | 0.4796 | 0.4843 | 0.4898 | 0.4903 |
| | BM25OKAPI | 0.3015 | 0.4095 | 0.4251 | 0.4346 | 0.4395 | 0.4400 |
| | SPLADE | 0.0388 | 0.0542 | 0.0624 | 0.0715 | 0.0842 | 0.0866 |
| **Dense** | SBERT | 0.1522 | 0.2075 | 0.2187 | 0.2265 | 0.2307 | 0.2314 |
| | ANCE | 0.1791 | 0.2498 | 0.2674 | 0.2802 | 0.2919 | 0.2931 |
| | ARCTIC | 0.3164 | 0.4114 | 0.4255 | 0.4362 | 0.4439 | 0.4452 |
| | GRANITE | 0.3134 | 0.4144 | 0.4338 | 0.4458 | 0.4543 | 0.4551 |
| | OPENAI-3-SMALL | 0.3164 | 0.4075 | 0.4236 | 0.4356 | 0.4443 | 0.4453 |
| | OPENAI-3-LARGE | 0.2866 | 0.3726 | 0.3883 | 0.4016 | 0.4120 | 0.4130 |
| | OPENAI-ADA-002 | 0.2985 | 0.3905 | 0.4123 | 0.4255 | 0.4339 | 0.4350 |
| | GEMINI-EMBEDDING-001 | 0.3403 | 0.4463 | 0.4651 | 0.4776 | 0.4865 | 0.4871 |
| **Hybrid** | BM25PLUS_CE | 0.3821 | 0.4945 | 0.5136 | 0.5191 | 0.5217 | 0.5219 |
| | BM25OKAPI_CE | **0.3851** | **0.4965** | **0.5140** | **0.5202** | **0.5229** | **0.5231** |

Table 12: Additional retrieval performance on *WeatherArchive-Retrieval* across sparse, dense, and re-ranking models. Recall@k results (k=1,3,5,10,50,100).

| Category | Model | Recall@1 | Recall@3 | Recall@5 | Recall@10 | Recall@50 | Recall@100 |
|---|---|---|---|---|---|---|---|
| **Sparse** | BM25PLUS | 0.3701 | 0.5851 | 0.6418 | 0.6776 | 0.7910 | 0.8269 |
| | BM25OKAPI | 0.3015 | 0.5433 | 0.6119 | 0.6776 | 0.7940 | 0.8299 |
| | SPLADE | 0.0388 | 0.0746 | 0.1134 | 0.1821 | 0.4776 | 0.6448 |
| **Dense** | SBERT | 0.1522 | 0.2896 | 0.3403 | 0.4000 | 0.5015 | 0.5522 |
| | ANCE | 0.1791 | 0.3403 | 0.4209 | 0.5224 | 0.7791 | 0.8657 |
| | ARCTIC | 0.3164 | 0.5343 | 0.5940 | 0.6746 | 0.8209 | 0.9104 |
| | GRANITE | 0.3134 | 0.5463 | 0.6299 | 0.7194 | 0.8866 | 0.9463 |
| | OPENAI-3-SMALL | 0.3164 | 0.5164 | 0.5851 | 0.6776 | 0.8537 | 0.9194 |
| | OPENAI-3-LARGE | 0.2866 | 0.4806 | 0.5493 | 0.6507 | 0.8537 | 0.9224 |
| | OPENAI-ADA-002 | 0.2985 | 0.5104 | 0.6030 | 0.7015 | 0.8806 | 0.9552 |
| | GEMINI-EMBEDDING-001 | 0.3403 | 0.5731 | 0.6537 | 0.7493 | **0.9164** | **0.9582** |
| **Hybrid** | BM25PLUS_CE | 0.3821 | **0.6388** | **0.7224** | **0.7612** | 0.8119 | 0.8269 |
| | BM25OKAPI_CE | **0.3851** | **0.6388** | 0.7164 | **0.7612** | 0.8179 | 0.8299 |

Table 13: Additional retrieval performance on *WeatherArchive-Retrieval* across sparse, dense, and re-ranking models. NDCG@k results (k=1,3,5,10,50,100).

| Category | Model | NDCG@1 | NDCG@3 | NDCG@5 | NDCG@10 | NDCG@50 | NDCG@100 |
|---|---|---|---|---|---|---|---|
| **Sparse** | BM25PLUS | 0.3701 | 0.4968 | 0.5205 | 0.5320 | 0.5573 | 0.5631 |
| | BM25OKAPI | 0.3015 | 0.4439 | 0.4721 | 0.4941 | 0.5190 | 0.5247 |
| | SPLADE | 0.0388 | 0.0595 | 0.0749 | 0.0970 | 0.1604 | 0.1875 |
| **Dense** | SBERT | 0.1522 | 0.2283 | 0.2489 | 0.2680 | 0.2897 | 0.2978 |
| | ANCE | 0.1791 | 0.2730 | 0.3055 | 0.3376 | 0.3937 | 0.4077 |
| | ARCTIC | 0.3164 | 0.4430 | 0.4679 | 0.4939 | 0.5274 | 0.5418 |
| | GRANITE | 0.3134 | 0.4482 | 0.4829 | 0.5119 | 0.5496 | 0.5592 |
| | OPENAI-3-SMALL | 0.3164 | 0.4356 | 0.4642 | 0.4938 | 0.5334 | 0.5441 |
| | OPENAI-3-LARGE | 0.2866 | 0.4004 | 0.4286 | 0.4613 | 0.5072 | 0.5183 |
| | OPENAI-ADA-002 | 0.2985 | 0.4213 | 0.4600 | 0.4918 | 0.5312 | 0.5434 |
| | GEMINI-EMBEDDING-001 | 0.3403 | 0.4790 | 0.5125 | 0.5432 | 0.5816 | 0.5884 |
| **Hybrid** | BM25PLUS_CE | 0.3821 | 0.5316 | **0.5660** | 0.5788 | 0.5904 | 0.5928 |
| | BM25OKAPI_CE | **0.3851** | **0.5330** | 0.5648 | **0.5795** | **0.5921** | **0.5940** |

### K.3 WEATHER UNDERSTANDING EVALUATION

Table 14: Recall evaluation results on Vulnerability and Resilience dimensions across models (average row in gray, Average column in gray).

| Model | Vulnerability | | | Resilience | | | Average |
|---|---|---|---|---|---|---|---|
| | Exposure | Sensitivity | Adaptability | Temporal | Functional | Spatial | |
| GPT-4O | 0.654 | 0.679 | 0.557 | 0.675 | **0.675** | 0.524 | 0.627 |
| GPT-3.5-TURBO | 0.629 | 0.460 | 0.504 | 0.593 | 0.354 | 0.482 | 0.504 |
| CLAUDE-OPUS-4-1 | **0.841** | **0.859** | **0.766** | **0.862** | 0.636 | **0.672** | **0.773** |
| CLAUDE-SONNET-4 | **0.841** | 0.652 | 0.719 | 0.786 | 0.631 | 0.663 | 0.715 |
| GEMINI-2.5-PRO | 0.806 | 0.672 | 0.722 | 0.786 | 0.631 | 0.652 | 0.711 |
| DEEPSEEK-V3-671B | 0.764 | 0.460 | 0.745 | 0.848 | 0.598 | 0.605 | 0.670 |
| MINISTRAL-8B-IT | 0.444 | 0.217 | 0.268 | 0.431 | 0.406 | 0.372 | 0.356 |
| MIXTRAL-8X7B-IT | 0.266 | 0.226 | 0.290 | 0.300 | 0.190 | 0.305 | 0.263 |
| QWEN2.5-72B-IT | 0.792 | 0.551 | 0.643 | 0.767 | 0.531 | 0.508 | 0.632 |
| QWEN2.5-32B-IT | 0.532 | 0.396 | 0.465 | 0.653 | 0.446 | 0.404 | 0.483 |
| QWEN2.5-14B-IT | 0.430 | 0.466 | 0.300 | 0.414 | 0.240 | 0.331 | 0.363 |
| QWEN2.5-7B-IT | 0.384 | 0.168 | 0.313 | 0.381 | 0.281 | 0.355 | 0.314 |
| LLAMA-3.3-70B-IT | 0.389 | 0.503 | 0.287 | 0.503 | 0.537 | 0.351 | 0.428 |
| LLAMA-3-8B-IT | 0.241 | 0.175 | 0.153 | 0.238 | 0.253 | 0.281 | 0.224 |
| QWEN-3-30B | 0.719 | 0.428 | 0.394 | 0.814 | 0.354 | 0.416 | 0.521 |
| QWEN-3-4B | 0.370 | 0.261 | 0.201 | 0.527 | 0.675 | 0.266 | 0.383 |
| Average | 0.572 | 0.463 | 0.481 | 0.588 | 0.459 | 0.465 | 0.514 |

Figure 6: Comparison of LLM free-form QA performance across ROUGE-1 and LLM-Judge metrics

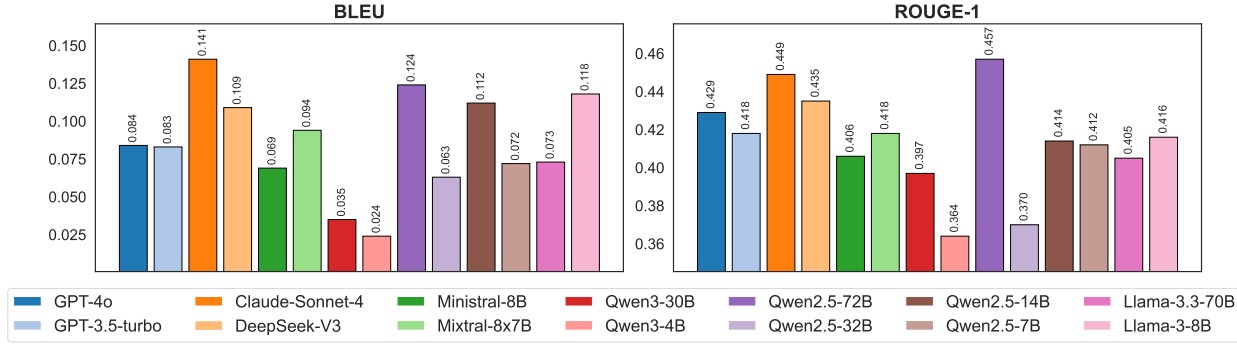

## L CASE STUDY

### L.1 SNOWSTORM

In this example, the ground truth reveals the insights into infrastructure and financial losses resulting from a snowstorm. The output of most LLMs is the same as the ground truth. However, *Qwen-72B* labeled the sensitivity as *moderate* instead of *critical*.

### L.2 RAINSTORM

In this example, the ground truth answer emphasizes the profound impact on real estate speculators and specific merchants caused by a record-breaking rainstorm. Most LLMs are able to provide similar answers to the ground truth, while they may underestimate the extent of the disaster and label the functional as *transportation*.

**Question:** How did the record-breaking rainstorm in Toronto impact the financial situation of real estate speculators and specific merchants, and what were the estimated monetary losses they incurred according to the passage?

**Reference Answer:** GOSSIP FROM TORONTO: A Record-Breaking Rainstorm Blared Real Estate Speculators Tided Over Their Trouble From our own correspondent Toronto, June 20 The rainstorm which deluged the city last evening was of the record-breaking character Nothing like it has been seen for some time The damage resulting is roughly estimated in the neighborhood of $100,000 Merchants in the central portion of the city were the greatest sufferers and Guinane's boot store and Armson & Stone's millinery emporium lose between them $2,000 on Yonge street The shopkeepers on the south side of King street, between Yonge and Bay streets, are very heavy losers, Caldwell & Hodgins, wine merchants, at Queen and John streets, will probably be minus $5,000 The affairs of the big speculators in real estate here, reported some time ago in difficulty, have been settled so as to avoid an assignment One prominent and wealthy lawyer, who is very largely interested in real estate in and about Toronto, has obtained a five years' extension from banks on $800,000, on which he will have to pay an interest of $48.

**Ground Truth**
**region**: Toronto, **exposure**: Sudden-Onset, **sensitivity**: Critical, **adaptability**: Constrained
**temporal**: short-term absorptive capacity, **functional**: information, **spatial**: local

**gpt-4o result**
**region**: Toronto, **exposure**: Sudden-Onset, **sensitivity**: Moderate, **adaptability**: Constrained
**temporal**: short-term absorptive capacity, **functional**: information, **spatial**: local

**Qwen-72B result**
**region**: Toronto, **exposure**: Sudden-Onset, **sensitivity**: Critical, **adaptability**: Constrained
**temporal**: short-term absorptive capacity, **functional**: transportation, **spatial**: local

## L.3  INTERNATIONAL SCOPE OF COVERAGE

The organization is based in Southern Quebec, yet it is an international news publisher. Therefore, the dataset is not geographically restricted to Quebec; it regularly includes reporting from the United States, Europe, South America, and other regions. This substantially broadens the dataset's geographic and topical diversity.

**United Kingdom**: STORMY WEATHER Heavy gales over the **United Kingdom** ... Disastrous loss of cattle shipments ... The British steamer Canopus, which arrived at **Liverpool yesterday from Boston**, lost her boats and 247 head of cattle due to heavy weather.

**United States**: **Nebraska, Omaha, Neb, February 12, Nebraska** is snowbound, For the past twenty-four hours a terrific blizzard has prevailed throughout the state, . . . , **Ohio, CLEVELAND**, February 12, A severe wind and snow storm from the northwest struck this city this morning, Nearly all trains are late, Street car traffic is almost entirely suspended.

**United States**: Brakeman Johnson, of freight engine No. 28, and Engineer Samuel Stowell, of engine No. 28, were killed, **Missouri, St. Louis, Mo**, February 12, Without warning from the weather . . . **Chicago** being eight hours behind time, The snow is four inches deep, which is phenomenal for this latitude, as time passed the storm increased in severity and at 2 p.m.

**TURKEY**, Bwsaltus snnwItloMs mt war, BrcnADHT, June 22, Russia has presented to Bulgaria another warship, also 16,000 rifles, Al baaia and rientenejrro, CossTAjmyoPL, June 22, The Porte declines to force the Albanians to surrender their territory to Montenegro, but is willing to use its persuasion, ARSTRIA-nTXGART, Klnteterial crisis, Yuxxa, June 22, A Ministerial crisis is imminent GEKJIAJY, ttiaaatrsnsi raiasu B kr Us, June 22, In the **district of Lambar, in Breslau, Prussia**, heavy torrential rains have killed 36 persons and destroyed 105 houses.

## M  THE USE OF LARGE LANGUAGE MODELS

This research study employed LLMs in two ways:

**Data processing and quality assurance.**    LLMs were used for post-OCR correction of the historical weather archive corpus. Given the multi-decade scale of digitized reports, this stage required approximately 0.3 billion input and output tokens to ensure accuracy and readability. OCR correction was necessary, as historical sources often contained scanning artifacts, unclear typography, and degraded text that could impair retrieval.

**Benchmark creation and evaluation.**    LLMs generate oracle annotations for the *WeatherArchive-Assessment* benchmark following the protocols described in Appendix G.1. We also include generative AI models as testbed models, including both proprietary and open-source LLMs, to evaluate their performance on climate-related downstream tasks in specialized domains.

All research design, data collection, and analysis were conducted solely by the authors without LLM's assistance.

