# OpenReview forum: "WeatherArchive-Bench: Benchmarking Retrieval-Augmented Reasoning for Historical Weather Archives"
_ICLR.cc/2026/Conference — Submitted to ICLR 2026_

### Official Review · Reviewer_vqED · 2025-10-26

**Soundness:** 2
**Presentation:** 3
**Contribution:** 3
**Rating:** 4
**Confidence:** 4

**Summary:**

The paper introduces WeatherArchive-Bench, a benchmark for RAG over historical weather archives. It contains two tasks: WeatherArchive-Retrieval, which deals with finding relevant passages from ~1M documents and WeatherArchive-Assessment which deals with classifying vulnerability/resilience indicators and answering free-form climate questions using retrieved passages.

**Strengths:**

1. The authors tackle an interesting and useful problem. Their dataset extracts societal vulnerability and resilience insights from historical weather narratives rather than only from meteorological data, which is an interesting perspective.

2. The corpus construction (OCR, GPT-4o post-correction) is carefully described and supported with quality checks, producing over a million passages. I think this will be a useful resource for the community.

3. The insight about how sparse retrieval might work better than dense retrieval on this domain is quite interesting, and might be more broadly useful for practioners from other fields who would want to design better retrieval systems for scientific applications.

**Weaknesses:**

1. The experimental setup is not described very clearly. The paper lacks critical details about the WeatherArchive-Assessment Task. First of all, Section 3.2.2 does not provide enough information about the experimental setup except to refer to the prompts in the Appendix and Figure 2. While the figure is useful, an explanation of how the dataset is constructed is required in the main text.

In addition, the experimental setup is also unclear. Is it retrieval-augmented QA (retrieve then answer) or direct QA given gold documents? If it's RAG-based, how is retrieval performed? Which retrieval model is used? Using the question directly, or through some query reformulation?


2. I am also not convinced by the choice of evaluation metrics for the Assessment Task. First of all, Table 3 reports results but doesn't clearly specify what metric is being shown

Moreover, in the freeform QA task, the use of token-level F1 for evaluating climate reasoning is inadequately justified. Token overlap is a surface-level metric that doesn't capture semantic equivalence or factual correctness. Climate-related answers often require domain expertise to verify - simple token matching may reward verbatim reproduction while penalizing valid paraphrases. The paper acknowledges this limitation by mentioning GPT-4.1-based evaluation: "Additionally, we employ LLM-based judgment using GPT-4.1 to evaluate climate reasoning quality beyond traditional similarity metrics, determining whether model-generated responses contain factual errors." However, I cannot locate these results in the paper. Can the authors point me to these results? They should be prominently featured given their importance for validating factual accuracy.

3. I also have concerns about the ground truth generation that require some clarification. Ground truth answers are generated from an LLM "using the rubric defined in Appendix G.1.1" from gold-standard documents. Test models are then evaluated against these LLM-generated answers. This creates a potentially circular evaluation. If the setup is retrieval + QA, the evaluation primarily measures whether the retrieval system can find the gold document. Once the correct document is retrieved, we would expect a reasonable LLM to produce answers similar to the LLM-generated ground truth. This would not test genuine climate reasoning ability, just retrieval effectiveness, and whether or not the target LLM produces a similar answer to the oracle LLM.

**Questions:**

Apart from the concerns raised in the weaknesses section, I have some minor suggestions/questions.

1. How many examples are in the WeatherArchive-Assessment Task? Is it one question per gold-standard report? This is not specified anywhere in the paper as far as I could tell.

2. In the introduction: "As shown in Table 1, existing benchmarks focus on relatively small-scale and primarily target scientific papers and reports rather than historically grounded archival data” - small scale what?

3. Figure 2 has a minor type: Anwser -> Answer.

---

> ### Author Response · Authors · 2025-11-21
> **Response to Reviewer vqED (Part1/2)**
>
> Dear Reviewer vqED,
>
> Thank you for these helpful comments, and we greatly appreciate the time and effort you have invested. We have revised our paper accordingly and highlighted the modifications in **Purple**.
>
> ### **Response to W1.1**:
> > The experimental setup is not described very clearly. The paper lacks critical details about the WeatherArchive-Assessment Task. First of all, Section 3.2.2 does not provide enough information about the experimental setup except to refer to the prompts in the Appendix and Figure 2. While the figure is useful, an explanation of how the dataset is constructed is required in the main text.
>
> We acknowledge that the description of the WeatherArchive-Assessment task can be clearer in the main text. The dataset is built by pairing historical newspaper narratives with daily weather records from the same location and time period. Each example includes a climate-related query, the corresponding retrieved passages from the archive, and a gold answer that reflects the documented weather impact or event. The newspaper articles are first digitized, cleaned, and aligned with structured meteorological data so that the narrative evidence and the weather report describe the same day or episode. We then create questions that require the model to link the retrieved narrative with the weather record, and we evaluate the model’s answer according to these aligned labels. In the revision, we will include a short and precise summary of this construction process directly so that readers can understand the task without relying on the appendix or the figure.
>
> ### **Response to W1.2**:
> > In addition, the experimental setup is also unclear. Is it retrieval-augmented QA (retrieve then answer) or direct QA given gold documents? If it's RAG-based, how is retrieval performed? Which retrieval model is used? Using the question directly, or through some query reformulation?
>
> Our setup follows a retrieve-then-answer pipeline. The retriever runs first and selects the top passages, and the QA model generates its answer only from what is retrieved. Retrieval is done with BM25 in our manuscript, using the query exactly as written, since all passages in the archive are already chunked. The generation step then takes these retrieved passages as its evidence and produces the final answer. We will clarify this flow more clearly in Section 3.2.2, therefore, the end-to-end setup is easy to follow.
>
> ### **Response to W2.1**:
> > I am also not convinced by the choice of evaluation metrics for the Assessment Task. First of all, Table 3 reports results but doesn't clearly specify what metric is being shown
>
> Thanks for catching this. The metric reported in Table 3 is the F1 score, and we will label this in the revised manuscript.
>
> ### **Response to W2.2**:
> > Moreover, in the freeform QA task, the use of token-level F1 for evaluating climate reasoning is inadequately justified. Token overlap is a surface-level metric that doesn't capture semantic equivalence or factual correctness. Climate-related answers often require domain expertise to verify - simple token matching may reward verbatim reproduction while penalizing valid paraphrases.
>
> Token-level F1 is a standard metric in free-form QA [1,2,3]. To address its limits in capturing semantic equivalence, we also did report semantic-level metrics (e.g., BERTScore) and an LLM-as-a-judge evaluation in Figure 5, which together provide a more solid assessment of climate-reasoning quality.
>
>
> ### **Response to W2.3**:
>
> > The paper acknowledges this limitation by mentioning GPT-4.1-based evaluation: "Additionally, we employ LLM-based judgment using GPT-4.1 to evaluate climate reasoning quality beyond traditional similarity metrics, determining whether model-generated responses contain factual errors." However, I cannot locate these results in the paper. Can the authors point me to these results? They should be prominently featured given their importance for validating factual accuracy.
>
> Thank you for this excellent suggestion. The results are shown in Appendix K (Figure 5). While these results are currently in the appendix, we agree they are important for validating factual accuracy and will move them to the main body in this revision.
>
> ## Reference
> [1] Bonomo et al. "LiteraryQA: Towards Effective Evaluation of Long-document Narrative QA." EMNLP. 2025.
>
> [2] Lee, et al. "KOBLEX: Open Legal Question Answering with Multi-hop Reasoning." EMNLP. 2025.
>
> [3] Xu, Shicheng, et al. "A Theory for Token-Level Harmonization in Retrieval-Augmented Generation." ICLR 2025.

---

> ### Author Response · Authors · 2025-11-21
> **Response to Reviewer vqED (Part2/2)**
>
> ### **Response to W3**:
>
> > I also have concerns about the ground truth generation that require some clarification. Ground truth answers are generated from an LLM "using the rubric defined in Appendix G.1.1" from gold-standard documents. Test models are then evaluated against these LLM-generated answers. This creates a potentially circular evaluation. If the setup is retrieval + QA, the evaluation primarily measures whether the retrieval system can find the gold document. Once the correct document is retrieved, we would expect a reasonable LLM to produce answers similar to the LLM-generated ground truth. This would not test genuine climate reasoning ability, just retrieval effectiveness, and whether or not the target LLM produces a similar answer to the oracle LLM.
>
> Thanks for raising the concern. We want to clarify that large-scale human-written answers are prohibitively expensive, while we want to keep ecological diversity. We use GPT-4.1 as the oracle because it achieves the highest preliminary testing with human judgments, and all generated answers undergo human verification.
>
> Also, the oracle in WeatherArchive-Assessment does not simply restate retrieved content. It follows a theory-driven rubric grounded in prior climate-vulnerability and resilience literature, which emphasizes conceptual criteria, such as causal drivers, exposure conditions, and vulnerability mechanisms. We acknowledge that LLM-derived labels introduce potential bias; the combination of human verification and rubric-based answer generation substantially mitigates this concern and provides a reliable, scalable basis for evaluating model performance.  We believe WeatherArchive-Bench represents the **first evaluation benchmark in the weather-archival domain**, serving as a crucial first step to illustrate how societies have experienced and responded to extreme weather events and to guide future research in climate impact and adaptation studies.
>
> ### **Response to Q1**:
> > How many examples are in the WeatherArchive-Assessment Task? Is it one question per gold-standard report? This is not specified anywhere in the paper as far as I could tell.
>
> There are 335 examples in the WeatherArchive-Assessment Task (Line 211), with a one-to-one mapping between queries and reports, resulting in a total of 335 query-answer pairs.
>
> ### **Response to Q2**:
> > In the introduction: "As shown in Table 1, existing benchmarks focus on relatively small-scale and primarily target scientific papers and reports rather than historically grounded archival data” - small scale what?
>
> “small-scale” paper source. Thank you for pointing this out! We will correct the typo in this revision.
>
> ### **Response to Q3**:
> > Figure 2 has a minor type: Anwser -> Answer.
>
> Thank you for pointing this out! We will correct typo in this revision.
>
>
> We sincerely thank you again for your insightful and constructive feedback on our work. We hope these clarifications and improvements further strengthen the paper. If you have any remaining questions or require further details, we would be very happy to provide them.
>
> Best wishes,
>
> The Authors

---

> ### Author Response · Authors · 2025-11-25
> **Rebuttal Follow-up**
>
> Dear reviewer,
>
> We hope the above clarifications in the revised paper sufficiently address your concerns. If it meets your expectations, we kindly ask you to consider reassessing the score to reflect the newly added discussion. We remain committed to addressing any remaining points you may have during the discussion phase!
>
> Thanks.

---

### Official Review · Reviewer_P5uj · 2025-11-01

**Soundness:** 3
**Presentation:** 3
**Contribution:** 3
**Rating:** 6
**Confidence:** 3

**Summary:**

This paper introduces WEATHERARCHIVE-BENCH, the first large-scale benchmark for evaluating retrieval-augmented reasoning (RAG) systems on historical weather archives. These archives capture how societies experienced and responded to extreme weather events, offering unique insights into societal vulnerability and resilience often absent from meteorological records. The benchmark includes two tasks: WeatherArchive-Retrieval, which measures models’ ability to locate relevant archival passages, and WeatherArchive-Assessment, which evaluates large language models (LLMs) in classifying vulnerability and resilience indicators from historical narratives.

The paper’s main contributions are:

1. A new 1M-document benchmark combining retrieval and reasoning tasks for historical climate archives.

2. A curated, high-quality corpus with OCR correction and expert validation for climate research applications.

3. Comprehensive evaluation showing that sparse and hybrid retrievers outperform dense ones on archaic vocabulary, and that even frontier LLMs struggle to reason about complex socio-environmental systems.

**Strengths:**

1. The paper presents the first benchmark addressing retrieval-augmented reasoning on historical climate archives, filling a major evaluation gap in climate-focused NLP.
2. The method demonstrates high methodological quality through large-scale data curation, OCR correction, and expert validation ensuring dataset reliability.
3. Offers clear task design (retrieval and assessment) that mirrors real-world workflows of climate scientists.
4. Provides comprehensive experiments across diverse retrievers and LLMs, yielding valuable diagnostic insights into model limitations.
5. The paper is well-organized and clearly written.

**Weaknesses:**

1. The dataset’s geographical and temporal scope focuses mainly on Southern Quebec, which could introduce potential locational bias.
2. LLM judges could have high variability of their assessments, especially when asked to generate multiple metric scores at once (as in G1.1.1). One way to verify the credibility of these results is to either generate multiple LLM judge decisions for a QA pair, or asking the judge to evaluate 1 score at a time, then compare the difference with original prompt.
3. To show the robustness of the dataset, one suggestion is to have another small dataset from a different region and examine whether the models have similar trends in performance.

**Questions:**

Please see weaknesses.

---

> ### Author Response · Authors · 2025-11-21
> **Response to Reviewer P5uj**
>
> Dear Reviewer P5uj,
>
> Thank you for your careful review and insightful feedback. We greatly appreciate the time and effort you have invested. We have revised the paper accordingly and highlighted them in **Orange** for clarity.
>
> ### **Response to W1 and W3**:
> > The dataset’s geographical and temporal scope focuses mainly on Southern Quebec, which could introduce potential locational bias.
>
> > To show the robustness of the dataset, one suggestion is to have another small dataset from a different region and examine whether the models have similar trends in performance
>
> The organization is based in Southern Quebec, yet it is an international news publisher. Therefore, the dataset is not geographically restricted to Quebec; it regularly includes reporting from the United States, Europe, South America, and other regions. This substantially broadens the dataset’s geographic and topical diversity. Below are a few representative samples:
>
> #### *"ellipsis"*: ... indicates omitted content for brevity
> **United Kingdom**: ``STORMY WEATHER Heavy gales over the United Kingdom ... Disastrous loss of cattle shipments ... The British steamer Canopus, which arrived at **Liverpool yesterday from Boston**, lost her boats and 247 head of cattle due to heavy weather.``
>
> **United States**:  ``Nebraska, Omaha, Neb, February 12, Nebraska is snowbound, For the past twenty-four hours a terrific blizzard has prevailed throughout the state, …, Ohio, CLEVELAND, February 12, A severe wind and snow storm from the northwest struck this city this morning, Nearly all trains are late, Street car traffic is almost entirely suspended.``
>
> **United States**: ``Brakeman Johnson, of freight engine No. 28, and Engineer Samuel Stowell, of engine No. 28, were killed, Missouri, St. Louis, Mo*, February 12, Without warning from the weather … Chicago being eight hours behind time, The snow is four inches deep, which is phenomenal for this latitude, as time passed the storm increased in severity and at 2 p.m.``
>
> **Germany**: ``TURKEY, Bwsaltus snnwItloMs mt war, BrcnADHT, June 22, Russia has presented to Bulgaria another warship, also 16,000 rifles, Al baaia and rientenejrro, CossTAjmyoPL, June 22, The Porte declines to force the Albanians to surrender their territory to Montenegro, but is willing to use its persuasion, ARSTRIA-nTXGART, Klnteterial crisis, Yuxxa, June 22, A Ministerial crisis is imminent GEKJIAJY, ttiaaatrsnsi raiasu B kr Us, June 22, In the district of Lambar, in Breslau, Prussia, heavy torrential rains have killed 36 persons and destroyed 105 houses.``
>
> As suggested, we will include these samples in Appendix L as case studies in the revised manuscript. Thank you!
>
> ### **Response to W2**:
>
> > LLM judges could have high variability of their assessments, especially when asked to generate multiple metric scores at once (as in G1.1.1). One way to verify the credibility of these results is to either generate multiple LLM judge decisions for a QA pair, or asking the judge to evaluate 1 score at a time, then compare the difference with original prompt.
>
> We appreciate the reviewer’s concern about the potential variability of LLM-based judging. We agree that generating multiple independent LLM-judge decisions or prompting the judge for one score at a time could indeed improve robustness, yet doing so at scale is really expensive for our setting. To mitigate this limitation, we **human-verify LLM-generated oracle QA outputs** to ensure quality and consistency. Additionally, we **randomly sample 50 QA instances** (approximately one-seventh of the dataset) and obtain four independent human annotators to verify answer quality. We will clarify this trade-off in the revised manuscript and highlight the complementary role of human verification in reinforcing the credibility of our evaluation results.
>
> We hope these responses have adequately addressed your concerns. Let us know if there are still any concerns!
>
> Best wishes,
>
> The Authors

---

### Official Review · Reviewer_vWhU · 2025-11-01

**Soundness:** 3
**Presentation:** 2
**Contribution:** 2
**Rating:** 4
**Confidence:** 4

**Summary:**

The paper presents a new retrieval and LLM evaluation benchmark with a focus on historical weather reports.

## Retrieval benchmark
Data sources are news reports from Southern Quebec, dated either between 1880-1899 or 1995-2014.

They OCR the articles, use GPT-4 to fix the error, split them into 1M overlapping chunks of 256 tokens, use keyword search to select 525 passages, and manually select 335 from these.

They use an LLM to make queries from single gold passages resulting in a retrieval benchmark.

## Assessment Benchmark
The 335 passages are annotated with categorical values across six dimensions using prompting + human verification. There is also a QA part of the benchmark but it is not clear from the main text how the answers are made.

## Metrics
Eval is done with retrieval / classification metrics respectively, answers are judged on string similarity and with an LLM judge.

## Results
Experiments are done for a solid selection of retrieval models and LLM. Results look plausible yet unsurprising. There seems to be significant headroom left but it is not clear to what extend clever prompting can close that gap.

**Strengths:**

New benchmark with headroom

Currently under-evaluated domain

The language used in the reasoning / classification task is dated and comes from a noisy extraction process. This makes it an interesting task to measure resilience / stability of LLMs (less of an issue for dense retrieval)

Experiments on a wide range or retrievers / LLMs

Extensive Appendix

**Weaknesses:**

The work uses too much justification of the task and general descriptions on why the task is hard and relevant. That space could be used for a running example or deeper discussion of corpus curation and experimental results and would allow the work to rely less on the appendix.

Some statements are insufficiently grounded interpretations. For example they report 91% BLEU for the text extraction (OCR + GPT fixing) and go on to state:

```
These results confirm that the GPT-4o correction pipeline successfully preserves semantic content
while eliminating OCR artifacts that would otherwise compromise retrieval accuracy and down-
stream task performance.
```

This might well be true but there is no evidence for it or contrasting results either without the fixing part or with gold documents.

The prompt that makes queries from passages requests that the question should not be answerable from similar passages but that interpretation is left to the model. A stronger method could retrieve some documents for an initial query and include these distractors in the context.

Fixing text with GPT might introduce model bias and need to be acknowledged

**Questions:**

The most interesting part of this corpus is the shift in domain to somewhat outdated language. There are some example in the appendix but this could be explored further. Should the keywords be selected to match archaic language? Can the corpus be 'translated' to modern language use? Do models exhibit specific error pattern that are tied to shifts in language use?

Also teasing apart errors from language vs. the noisy nature of the data would be interesting to show what models are resilient to.

## Improvement Suggestions

* Consistent counting of corpus size: Sometimes the size of the weather bench corpus is counted in #papers (Table 1), then 'number of historical archives' (end of Introduction), passages (middle of Page 2), articles, chunks.

* How were the articles selected? Are they just full newspapers from that time? Are the old and new parts of similar size?

* The description of why the task is important and the data relevant is long, wordy, and subjective. Some examples would help

* Learning more about the pipeline would be interesting. How was the OCR done, some example of OCR errors and how they were corrected with GPT, some analysis of the remaining errors.

* In line 269 it says 'Adaptive capacity' but Figure 2 calls that dimension "Adaptability"

* I don't understand the paragraph in lines 296-302.

* Line 342: Proprietary, Gemini

* Table 3 would be easier to read with percentages or color coding

* Contrary to the text in 5.2, results in Table 3 are not much worse for Adaptability vs. Sensitivity.

---

> ### Author Response · Authors · 2025-11-21
> **Response to Reviewer vWhU (Part 1/3)**
>
> Dear Reviewer vWhU,
>
> Thank you so much for your helpful comments and efforts on our research work. We would like to clarify these concerns as follows and have revised the paper accordingly (changes are marked in **Green**).
>
> ### **W1**:
> > The work uses too much justification ... would allow the work to rely less on the appendix.
>
> Thank you for the suggestion. Our main paper already includes Figure 1 and Figure 2, which outline the full workflow for corpus curation and experimentation, and Appendix L provides case studies of the annotation outcomes. The reason why we put much effort into the task justification is that it is motivated by theory-grounded, informed by discussions with meteorological experts, and aims to highlight the importance of capturing human climate perception in archival news for future climate-impact and adaptation research. In response to your feedback, we are considering moving one of the case studies from the appendix into the main body.
>
> ### **W2**:
> > Some statements are insufficiently grounded interpretations. For example they report 91% BLEU for the text extraction ... This might well be true but there is no evidence for it or contrasting results either without the fixing part or with gold documents.
>
> We clarify that our post-OCR pipeline uses GPT-4o with task-specific prompts to restore damaged textual sequences while retaining the original semantic content. To evaluate the correction quality, we compare the corrected text against human-verified references using BLEU and ROUGE, obtaining high consistency scores. While we agree that these metrics alone do not fully establish causal improvements to downstream tasks, they do provide quantitative evidence that the corrected text more closely matches human-curated ground truth than the raw OCR output, as they provide lexical restoration [1] and structural normalization in reducing character error rates [2]. We have added concrete before-and-after examples here to illustrate the types of structural distortions that the correction module successfully resolves below. We will add samples for illustration to the appendix in this revision.
>
> OCR'd text:
> `OXTAEIO Ta os3cmo, Febrnarr 13, A ICy day, A collision oc-enrrel oc-enrrel on the Caada Simthern K iilwar, outit a mile and half wet of TUsonborg, t three 0 ‘dock this moruing, oetwaell twj t7 freight traios, coaaisting about 40 ar sach, mostlf ladett with ZG i-CraJ i-CraJ nercaaoxiuie, a4 low Wi ot pork, grain, tc... `
>
> Post-OCR correction:
> `OXTAEIO, Taos, February 13, A icy day, A collision occurred on the Canada Southern Railway, about a mile and a half west of Tusonborg, at three o'clock this morning, between two freight trains, consisting of about 40 cars each, mostly loaded with ZG i-CraJ merchandise, and low with pork, grain, etc...`
>
> ### **W3**:
> > The prompt that makes queries from passages requests ... A stronger method could retrieve some documents for an initial query and include these distractors in the context.
>
> We acknowledge that stronger methods could incorporate distractor documents into the context. However, we argue that our design follows an **end-to-end RAG pipeline**. The LLM-based generator relies on the top-3 BM25 passages, which often contain weak or even missing evidence. This setup is intentional, as it allows us to directly observe how retrieval errors propagate into the final answers. Indeed, many retrieved sets do not include the gold evidence at all.
>
> To clarify this point, we added experiments with higher-performing retrievers. As shown below, **better retrieval quality consistently yields better downstream QA performance**, confirming the expected trend:
>
> |Setting|BLEU|ROUGE-1|ROUGE-L|BERTScore|F1 Score|LLM Judge Score|
> |-|-|-|-------|---------|----------|----------------|
> |Qwen2.5-14B (ANCE)|0.0558|0.2859|0.2566|0.8555|0.2636|0.1015|
> |Qwen2.5-14B (BM25)|0.0746|0.3089|0.2758|0.8602|0.2919|0.1373|
> |Qwen2.5-14B (Gemini-Embedding)|0.0742|0.3234|0.2911|0.8668|0.2939|0.2030|
> |Qwen2.5-72B (ANCE)|0.0677|0.3113|0.2809|0.8615|0.2884|0.1612|
> |Qwen2.5-72B (BM25)|0.0817|0.3353|0.3001|0.8596|0.3065|0.1851|
> |Qwen2.5-72B (Gemini-Embedding)|0.0934|0.3574|0.3262|0.8719|0.3315|0.3224|
>
> ## Reference
>
> [1] Löfgren, et al. "Post-OCR correction of digitized Swedish newspapers with ByT5." LaTeCH-CLfL, ACL 2024.
>
> [2] Kanerva, et al. "OCR Error Post-Correction with LLMs in Historical Documents: No Free Lunches." RESOURCEFUL, ACL 2025.

---

> ### Author Response · Authors · 2025-11-21
> **Response to Reviewer vWhU (Part 2/3)**
>
> ### **W4**
> > Fixing text with GPT might introduce model bias and need to be acknowledged.
>
> Thanks for your comment. We will acknowledge this limitation in the revised manuscript. As noted in prior literature [2], employing an LLM for text fixing can introduce biases, but our use of GPT-4.1 at this stage reflects a practical choice that is common in data cleaning and text normalization workflows [1, 3]. Importantly, text correction is not the primary focus of our work. Our contribution centers on enabling climate-adaptation research by releasing an open-source, structured dataset derived from historical weather reports. The dataset and its construction pipeline can support downstream studies in information extraction, historical document analysis, and domain-specific retrieval, independent of the particular correction model used.
>
> ### **Q1**:
> > The most interesting part of this corpus ... Do models exhibit specific error patterns that are tied to shifts in language use?
>
> Thank you for the insightful question. We agree that the interaction between archaic language in archival news and modern model behaviour is an important direction. As a preliminary analysis, we tested how substituting an older term (e.g., “inundation”) with its modern equivalent (“flooding”) affects retrieval performance.
>
> As an example, we tested the following two queries, using **inundation** vs. **flooding** in a query. Recall@k for queries using older (`o_r`) vs. modern (`m_r`) weather terms.
>
> |retriever|o_r@1|o_r@3|o_r@5|o_r@10|m_r@1|m_r@3|m_r@5|m_r@10|
> |-|-|-|-|-|-|-|-|-|
> |BM25Plus|0|1|1|1|0|1|1|1|
> |BM25Okapi|0|1|1|1|0|1|1|1|
> |Splade|0|0|0|0|0|0|0|0|
> |Sbert|0|1|1|1|0|1|1|1|
> |Ance|1|1|1|1|0|0|0|1|
> |Arctic|1|1|1|1|1|1|1|1|
> |Granite|1|1|1|1|1|1|1|1|
>
> These results demonstrate that archaic vocabulary can indeed impact retrieval, with some models (e.g., ANCE) struggling to associate older terms with their modern equivalents. This suggests that linguistic drift in archival news can introduce systematic retrieval challenges, which will be an interesting avenue for future work.
>
> ### **Q2**:
> > Also teasing apart errors from language vs. the noisy nature of the data would be interesting to show what models are resilient to.
>
> Yes, to isolate language effects from OCR-induced noise as you suggested, we also evaluated retrieval on the pre-OCR dataset. The results are shown below. Sparse retrievers achieve the highest recall, indicating stronger robustness to noise, likely because keyword-based matching is less affected by OCR distortion. Dense models degrade more substantially relative to the post-OCR setting, suggesting greater sensitivity to noisy input. We will supplement these experiments in this revision.
>
> |retriever_type|r@3|r@10|r@50|r@100|
> |-|-|--|--|-|
> |BM25Plus|0.3224|0.4746|0.7403|0.8388|
> |BM25Okapi|0.3194|0.4806|0.7612|1.0000|
> |arctic|0.2478|0.3791|0.5373|0.5881|
> |ance|0.1254|0.2209|0.3910|0.4507|
> |sbert|0.1075|0.1791|0.2328|0.2866|
> |BM25L|0.0388|0.0836|0.1463|0.1821|
> |model_splade|0.0090|0.0418|0.1075|0.1433|
>
> ### **Q3**
> > Consistent counting of corpus size: Sometimes the size of the weather bench corpus is counted in #papers (Table 1), then 'number of historical archives' (end of Introduction), passages (middle of Page 2), articles, chunks.
>
> Thanks for pointing this out. The number of historical archives (# papers) should be 1,035,862 ≈ 1.04 M - we will correct this typo in this revision.
>
> ### **Q4**
> > How were the articles selected? Are they just full newspapers from that time? Are the old and new parts of similar size?
>
> Articles are selected based on keyword frequency search. We search for keywords like “snowstorm”, “flood”, etc (Table 5). Each hit is then manually confirmed to see whether it is actually related to a disaster. If it is, the article is added to the 335-line database.
> Yes, they are full newspapers of a certain date, but for our task, we first chunked them into 256-token segments with an overlap of 100 tokens.
>
> The tokenizer we used was `cl100k_base`. The ratio of new to old articles in the post-OCR database is `1,211,803 : 336,325`, and in the pre-OCR database, the ratio is `3,132,306 : 445,093`. We will supplement these discussions in the revision.
>
> ### **Q5**
> > The description of why the task is important and the data relevant is long, wordy, and subjective. Some examples would help.
>
> We appreciate this feedback. As noted in **W1**, we are considering moving one case study from the appendix into the main body to improve clarity and accessibility.
>
> ## Reference
>
> [1] Choi, et al. "Multi-news+: Cost-efficient dataset cleansing via llm-based data annotation." EMNLP 2024.
>
> [2] Muñoz-Ortiz, et al. "Contrasting linguistic patterns in human and LLM-generated news text." Artificial Intelligence Review. 2024
>
> [3] Naeem, et al. "RetClean: Retrieval-Based Data Cleaning Using LLMs and Data Lakes." VLDB 2024.

---

> ### Author Response · Authors · 2025-11-21
> **Response to Reviewer vWhU (Part 3/3)**
>
> ### **Q6**
> > Learning more about the pipeline would be interesting. How was the OCR done, some example of OCR errors and how they were corrected with GPT, some analysis of the remaining errors.
>
> We agree that more details on the preprocessing pipeline would help readers.
>
> **OCR Noise in Historical Newspapers**: Our corpus comes from historical newspaper scans that exhibit typical OCR noise—broken words, missing characters, and misread symbols—caused by mixed layouts and complex narrative structure, which reduces interpretability and limits suitability for downstream climate-impact analysis.
>
> **Post-OCR Correction Strategy**: We apply post-OCR correction targeting errors that impair human comprehension and downstream information extraction, including fragmented or incorrectly split words, spurious symbols from layout irregularities, and disrupted sentence structure. We then use OpenAI models to clean short OCR segments while preserving semantic content, and this procedure aligns with established post-OCR practices [1] for mitigating artifacts in documents with mixed content formats and complex narrative structure.
>
> We will supplement these discussions in the revision.
>
> ### **Q7**
> > In line 269 it says 'Adaptive capacity' but Figure 2 calls that dimension "Adaptability"
>
> Thanks for pointing this out. We will correct this typo.
>
> ### **Q8**
> > I don't understand the paragraph in lines 296-302.
>
> We realize that the paragraph may feel unclear because it refers to several parts of the pipeline without giving enough explanation. What we intended to describe is the full retrieval-to-generation process. Each climate query is first handled by the retriever, which selects the top three passages that form the evidence used by the model. The QA stage then takes these retrieved passages and produces an answer, and we compare this answer to the gold reference to check whether the pipeline works as a whole. The earlier classification tasks measure how well models extract structured signals, while the open-ended QA task measures whether the pipeline can combine information from different archival sources and describe the climate impact in a clear and scientifically useful way. In the revision, we will clarify these connections so the paragraph reads more directly.
>
> ### **Q9**
> > Line 342: Proprietary, Gemini.
>
> Thanks for pointing this out; we will correct the typo in this revision.
>
> ### **Q10**
> > Table 3 would be easier to read with percentages or color coding
>
> Thanks for your suggestion; we will transform them into percentages in this revision
>
> ### **Q11**
> > Contrary to the text in 5.2, results in Table 3 are not much worse for Adaptability vs. Sensitivity.
>
> Thanks! The wording in Section 5.2 may have implied that `Sensitivity` performance should be markedly worse than `Adaptability`, which was not our intention. The phrase “in contrast” was meant only to highlight that `Sensitivity` classification relies more heavily on reasoning, particularly on latent social and institutional factors discussed in Morss et al. (2011). We will revise the text to avoid this confusion and use more neutral framing (e.g., “on the other hand”). We appreciate the reviewer for catching this ambiguity, and we will update the manuscript accordingly.
>
> We sincerely thank all the reviewers for their valuable and constructive comments, which greatly help us improve the quality of our paper.
>
> Best wishes,
>
> The Authors
>
> ## Reference
>
> [1] Zhang, et al. "Post-OCR correction with OpenAI's GPT models on challenging English prosody texts." ACM DocEng 2024.

---

> ### Author Response · Authors · 2025-11-25
> **Rebuttal Follow-up**
>
> Dear reviewer,
>
> We hope the above clarifications and the additional experiments in the revised paper sufficiently addressed your concerns. If so, could you please consider updating the overall score. Besides, we are still pleased to address any other questions you may have during the discussion phase.
>
> Thanks.

---

### Official Review · Reviewer_CZQs · 2025-11-01

**Soundness:** 2
**Presentation:** 2
**Contribution:** 1
**Rating:** 2
**Confidence:** 2

**Summary:**

This paper introduces WEATHERARCHIVE-BENCH, a new benchmark for evaluating Retrieval-Augmented Generation (RAG) systems in the domain of historical climate analysis. The primary contribution is a novel, large-scale corpus of 1.05 million (1.05M) archival news segments derived from Southern Quebec archives spanning 1880-1899 and 1995-2014. The benchmark proposes two distinct tasks: 1. WeatherArchive-Retrieval: An information retrieval task that evaluates a system's ability to locate relevant passages from the 1.05M-passage corpus, using a test set of 335 curated passages and synthetically generated queries. 2. WeatherArchive-Assessment: A reasoning task that evaluates a large language model's (LLM) capacity to classify indicators of societal vulnerability (Exposure, Sensitivity, Adaptability) and resilience (Temporal, Functional, Spatial scales) from a given archival passage.

**Strengths:**

1. The paper's primary strength lies in its novel and significant problem formulation. The motivation to leverage "rich, untapped narratives" (Line 013) from historical archives to inform contemporary climate adaptation and disaster preparedness is both original and of high societal value. This interdisciplinary effort to bridge digital humanities, NLP, and climate science represents a valuable and under-explored research direction.

2. The most significant and durable contribution of this work is the curation and public release of the 1.05M-passage corpus (Contribution 2, Line 105). This large-scale dataset of "OCR-parsed archival documents" (Line 098)  spanning two distinct historical periods is a valuable new resource for the research community, independent of the benchmark's other components, and will undoubtedly enable future research in historical NLP and climate-related text mining.

3. A final strength is the clear and rigorous theoretical grounding of the "WeatherArchive-Assessment" task. The authors move beyond simple question-answering by building their evaluation on well-defined frameworks for societal vulnerability (O'Brien et al., 2004)  and resilience (Feldmeyer et al., 2019). By operationalising these established climate science concepts (detailed in Appendix E, Tables 7 and 8) , the paper provides a structured, nuanced schema for evaluating complex socio-environmental reasoning.

**Weaknesses:**

1. The paper's entire premise is built on a fundamental contradiction. It is motivated by the challenge of "noisy digitized quality" and "Optical Character Recognition (OCR) errors", yet Appendix A (Line 756) reveals that the entire corpus was pre-processed using GPT-4o to "systematically correct OCR errors" (Line 762)  before any evaluation. This benchmark does not, therefore, measure robustness to OCR noise, arguably the primary technical hurdle in this domain. It measures performance on an artificially clean dataset, rendering all findings un-generalizable to real-world archival data. This is a complete methodological misalignment, as modern benchmarks for this problem, such as OHRBench, are specifically designed to evaluate the cascading impact of OCR by systematically introducing and varying noise, not eliminating it entirely.

2. The WeatherArchive-Assessment task is built on a methodologically circular and tenuous foundation. The "ground-truth oracles" were generated by GPT-4.1, and other LLMs are then evaluated against these synthetic labels (Table 3). This is not a test of correctness but a test of model-agreement, measuring how well Claude-Opus can imitate the specific biases and outputs of GPT-4.1. The authors' attempt at validation in Appendix F is a major weakness, not a strength: a $\kappa_{Fleiss}$ of only 0.67 among human experts does not indicate a "gold standard" but rather proves the underlying task categories are ambiguous and subjective. The subsequent claim that GPT-4.1 "surpasses average human performance"  is a  misrepresentation of this ambiguous data.

3. The paper's central claim to be a "benchmark for evaluating retrieval-augmented generation (RAG) systems" (Line 018)  is false. The paper does not evaluate a single end-to-end RAG pipeline. It presents two disconnected components: retrieval (Task 1, Line 186) and generation-on-gold-passages (Task 2, Line 225). A true RAG evaluation, as exemplified by benchmarks like KILT, must measure the propagated effect of retrieval quality on generation quality. This benchmark's design is primitive as it cannot answer the most critical question: how does a poorly retrieved passage (from Task 1) impact the LLM's reasoning (in Task 2)? It fails to evaluate the "Augmented" part of RAG entirely.

**Questions:**

1. On the Contradiction of the "Noise" Premise: The paper is motivated by "noisy digitized quality" but Appendix A (Line 756)  states you corrected all OCR errors with GPT-4o as a preprocessing step. This eliminates the core challenge. To convert this weakness, you could re-run all experiments on the raw, uncorrected corpus. Or, following OHRBench, you could release versions of the benchmark with varying, controlled levels of OCR noise to actually measure systemic robustness?

2. On the Circularity and Ambiguity of the "Ground Truth": The Assessment ground truth is synthetic (GPT-4.1-generated)  and validated with a human Fleiss' κ of only 0.67, indicating high ambiguity. Given the known issues of circular evaluation, how can you be certain your benchmark measures reasoning rather than just imitation of GPT-4.1's biases? To strengthen this, would you consider replacing the 335 synthetic oracles with a fully human-adjudicated gold standard, even if it must be a smaller subset?

3. On the Lack of End-to-End RAG Evaluation: You claim this is a "RAG" benchmark, but Task 1 and Task 2 are evaluated in isolation. This is a component-level, not an end-to-end, evaluation. To convert this to a true RAG benchmark (like KILT ), can you provide results for an end-to-end setting? Specifically, what is the performance on Task 2 when the LLM is provided with the actual top-k retrieved passages from Task 1 (e.g., from BM25 vs. ANCE)?

---

> ### Author Response · Authors · 2025-11-21
> **Response to Reviewer CZQs (Part1/2)**
>
> Dear Reviewer CZQs
>
> Thank you for your thorough review and insightful feedback. We appreciate the opportunity to clarify our contributions and address your concerns. We supplement several experiments to address your concerns and have revised the paper accordingly (changes are marked in **Blue**)
>
> ### **Response to W1**:
>
> We clarify that our benchmark is not meant to test OCR robustness. Our goal is to evaluate how well retrievers and LLMs can find, understand, and reason over climate-related archival articles. Although OCR noise is a known challenge, it is not the focus of our tasks, and keeping the noise would mainly distort retrieval and reasoning results. For this reason, we apply post-OCR correction following the technique proposed by Zhang et al. (2024) [1], to keep the evaluation centered on climate-content understanding and cross-document reasoning.
>
> **Intellectual property issues**: We further clarify that the original archival newspaper articles are subject to copyright and cannot be redistributed. Beyond IP constraints, many articles contain personally identifiable information (e.g., names, addresses, phone numbers) that may surface through OCR, and releasing the raw text would raise privacy concerns. Unlike OCR-focused benchmarks such as OHRBench, our dataset is intentionally built to study climate-article comprehension rather than sensitivity to OCR errors, and we will add these clarifications more clearly in the revision.
>
> ### **Response to W2**:
>
> **ground-truth oracle**: Our intention is not to claim that GPT-4.1 provides an absolute ground truth, but to use it as an initial oracle in a domain where no established annotation protocol or large-scale dataset currently exists. Followed by existing literature in climate task design [2], we run preliminary testing and find that GPT-4.1 achieves the highest agreement with human judgments among all models we examine, which is why we adopt it as a preliminary annotator.
>
> **model agreement**: We note that model-generated labels inevitably carry bias, and this setup is common practice in under-annotated domains [3,4]. We also human-verify each output to ensure data quality.
>
> **Human agreement**: This value does not imply the task is invalid; instead, it reflects the inherent difficulty of reasoning about societal vulnerabilities in complex climate contexts. As discussed in prior literature on climate communication and disaster analysis (summarized in Appendix E), our goal is to formalize these dimensions into a theoretically grounded and empirically validated framework, not to claim perfect consensus among annotators.
>
> Thanks for pointing out the phrase “surpasses average human performance.” We agree that this wording may suggest an unwarranted normative comparison. To avoid this ambiguity, we revise the statement to “GPT-4.1 achieves higher agreement with the reference annotations than the average individual annotator.” In this sense, the alternative annotator experiment measures alignment with the annotation protocol, not superiority over human expertise.
>
>
> ### **Response to W3**:
>
> **End-to-end RAG**: We believe there is a misunderstanding about our setup, which is described in Section 3.2.2. Our benchmark does include an end-to-end RAG pipeline: every question in the Weather-Assessment task is answered using the passages retrieved in Weather-Retrieval. The generation stage is not fed gold passages; it uses the top-3 retrieved passages, which naturally include cases with weak or missing evidence. This design lets us observe how poorly retrieved passages affect the model’s final answer, and in many cases the retrieved passages do not contain the gold evidence at all. In those settings, the model must explicitly state that no specific information is available. To make this clearer, we supplement results across four settings by the Qwen-14B-Instruct model: QA_no_context (no retrieved passages), QA_top3 (top-3 BM25 passages), QA_top3+gold (top-3 plus the gold passage), and QA_gold (gold passage only).
>
> |Setting|BLEU|ROUGE-1|ROUGE-L|BERTScore|F1 Score|LLM Judge Score|
> |-|-|-|-|-|-|-|
> |QA_no_context|0.0182|0.1847|0.1680|0.8456|0.1199|0.0328|
> |rQA_top3|0.1116|0.3089|0.2758|0.8781|0.2919|0.1373|
> |QA_top3+gold|0.1212|0.4141|0.3797|0.8851|0.3800|0.7552|
> |QA_gold|0.1577|0.4662|0.4248|0.8963|0.4577|0.7881|
>
> These results help clarify how retrieval quality impacts the full retrieval-to-generation pipeline and make explicit the evaluation settings used in our benchmark.
>
> ## Reference
>
> [1] Zhang, et al. "Post-OCR correction with OpenAI's GPT models on challenging English prosody texts." ACM DocEng 2024.
>
> [2] Schimanski, et al. "ClimRetrieve: A Benchmarking Dataset for Information Retrieval from Corporate Climate Disclosures." EMNLP 2024.
>
> [3] Manivannan, et al. "ClimaQA: An Automated Evaluation Framework for Climate Question Answering Models." ICLR 2025.
>
> [4] Wang, et al. “A User-Centric Multi-Intent Benchmark for Evaluating Large Language Models.” EMNLP 2024.

---

> ### Author Response · Authors · 2025-11-21
> **Response to Reviewer CZQs (Part2/2)**
>
> ### **Response to Q1:**
>
> **OCR Noise Premise** Thank you for raising this point. We believe there is a misunderstanding of our motivation. Our benchmark is not driven by OCR noise; rather, it is designed to study retrieval and reasoning over archival weather reports, where the core challenge lies in understanding historical disaster narratives, not in handling OCR artifacts.
>
> As suggested, we conducted the retrieval experiment on the before-OCR dataset. As the table shows, these results demonstrate that OCR noise indeed affects retrieval quality. This demonstrates the necessity of post-OCR correction; we will include the noisy-corpus analysis in Appendix A in the revision so that readers can better understand robustness under degraded inputs.
>
> |retriever_type|ndcg@3|ndcg@10|ndcg@50|ndcg@100|
> |--------------|------|-------|-------|--------|
> |raw_BM25Plus|0.2592|0.3152|0.3755|0.3917|
> |raw_BM25Okapi|0.2645|0.3230|0.3851|0.4237|
> |raw_arctic|0.2038|0.2514|0.2863|0.2946|
> |raw_ance|0.1035|0.1385|0.1753|0.1850|
> |raw_sbert|0.0860|0.1104|0.1221|0.1309|
> |raw_BM25L|0.0258|0.0421|0.0555|0.0613|
> |raw_splade|0.0068|0.0188|0.0336|0.0393|
>
>
> ### **Response to Q2**
>
> **Theory-Driven Annotations**: Thank you for raising this concern. We agree that synthetic references may introduce bias, and we acknowledge the ambiguity. Our benchmark, however, is designed to assess structural and conceptual reasoning over archival weather reports. The annotation protocol draws directly from climate-vulnerability theory, which constrains the influence of model-specific preferences and focuses evaluation on identifying causal drivers, exposure conditions, and vulnerability mechanisms rather than mirroring GPT-4.1’s stylistic tendencies.
>
> **Human-Verified Subset**: We clarify that our current methodology already includes multiple layers of human oversight designed to mitigate circularity. Every synthetic answer is **individually human-verified** to ensure alignment with the theory-driven rubric, and a **50-sample subset** is annotated by four independent annotators to assess the validity and consistency of the underlying categories. This mixed pipeline allows us to maintain broad coverage of archival reports, yet we acknowledge introducing a smaller, expert-curated subset could further strengthen the benchmark, and we plan to explore this as a complementary resource in future work.
>
> ### **Response to Q3**
>
> As noted in our response to W3, WeatherArchive-Assessment QA uses the top-k passages retrieved in WeatherArchive-Retrieve.  Each question is answered using the **actual top-k passages** retrieved in WeatherArchive-Retrieve, **not gold** passages. It receives the actual retrieved passages from BM25. Figures 3 and 5 in our manuscript report the generation performance under exactly this setting, which reflects how retrieval quality affects the final answers.
>
> To further address Q3, we provide an end-to-end QA comparison using top-3 passages retrieved by BM25, ANCE, and Gemini, each answered by Qwen2.5-14B and Qwen2.5-72B:
>
> |Setting|BLEU|ROUGE-1|ROUGE-L|BERTScore|F1 Score|LLM Judge Score|
> |-------|-----|-------|-------|---------|---|----------------|
> |Qwen2.5-14B (ANCE)|0.0558|0.2859|0.2566|0.8555|0.2636|0.1015|
> |Qwen2.5-14B (BM25)|0.0746|0.3089|0.2758|0.8602|0.2919|0.1373|
> |Qwen2.5-14B (Gemini-Embedding)|0.0742|0.3234|0.2911|0.8668|0.2939|0.2030|
> |Qwen2.5-72B (ANCE)|0.0677|0.3113|0.2809|0.8615|0.2884|0.1612|
> |Qwen2.5-72B (BM25)|0.0817|0.3353|0.3001|0.8596|0.3065|0.1851|
> |Qwen2.5-72B (Gemini-Embedding)|0.0934|0.3574|0.3262|0.8719|0.3315|0.3224|
>
> As shown in the table, **stronger retrieval quality consistently leads to better downstream QA performance** for both Qwen2.5-14B and Qwen2.5-72B. This aligns with our expectation that improved retrieval yields more accurate augmented generation, further demonstrating the robustness of our end-to-end RAG pipeline.
>
> Finally, we will state this more clearly so readers can see that the benchmark already evaluates the full RAG pipeline in this revision.
>
> We hope these responses have adequately addressed your concerns. Let us know if there are still any concerns!
>
> Best wishes,
>
> The Authors

---

> ### Author Response · Authors · 2025-11-25
> **Rebuttal Follow-up**
>
> Dear reviewer,
>
> We hope that the clarifications above and the additional experiments included in the revised version have addressed your concerns. If the revisions meet your expectations, we would appreciate your consideration of updating the score to reflect the new results and discussion. Please feel free to let us know if any points remain unclear!
>
> Thanks.

---

### Author Response · Authors · 2025-12-01
**Global Summary and Response to Area Chairs and Reviewers**

Dear Area Chairs and Reviewers,

We are very grateful for the reviewers' recognition of our work, such as:

**Reviewer CZQs**
> This large-scale dataset is a valuable new resource for the research community, independent of the benchmark's other components, and will undoubtedly enable future research in historical NLP and climate-related text mining.

**Reviewer vWhU**
> The language used in the reasoning / classification task is dated and comes from a noisy extraction process. This makes it an interesting task to measure resilience / stability of LLMs

**Reviewer P5uj**
> Offers clear task design (retrieval and assessment) that mirrors real-world workflows of climate scientists.

**Reviewer vqED**
> more broadly useful for practioners from other fields who would want to design better retrieval systems for scientific applications.

We have carefully addressed all the reviewers' concerns point by point, with extra experiments correspondingly provided. The core contributions, responses to common issues, and paper supplements are as follows:

## Core Contributions

**Climate Reasoning Bench.** We introduce WeatherArchive-Bench, the first benchmark for evaluating end-to-end retrieval-augmented generation systems that extract societal vulnerability and resilience insights from historical weather narratives within newspapers rather than relying solely on meteorological data.

**Large-Scale Dataset.** We release over one million OCR-processed and cleaned historical news segments, providing an unprecedented corpus for studying long-term patterns of climate impacts and societal responses. This resource enables future work in historical NLP, climate-focused text mining, and the development of systems that learn from past adaptation strategies.

**Empirical Insights.** Extensive experiments show that (1) sparse retrieval models achieve strong top-rank relevance on climate archives due to their robustness to domain-specific terminology, (2) state-of-the-art LLMs struggle with socio-environmental system effects, performing well on explicit damage indicators but failing to reason about cross-system dependencies and multi-scale resilience dynamics, and (3) although larger models improve lexical fidelity and semantic coherence in climate-specific QA, generating scientifically accurate answers from historical narratives remains challenging even for frontier systems.

## Responses to Common Issues
- **End-to-End RAG Evaluation.** WeatherArchive-Bench **explicitly evaluates full retrieval-to-generation performance**, and we provide additional experiments that clearly demonstrate this end-to-end behavior. These results confirm that the benchmark measures integrated RAG reasoning rather than isolated components.

- **OCR Correction.** Our OCR correction step ensures the benchmark **evaluates climate reasoning rather than noise robustness**. This choice keeps the focus on narrative understanding, which is central to our research goal.

- **Ground-Truth Oracle.** We use a GPT-4.1–based oracle **paired with human verification and a theory-grounded rubric**, this step is to confirm consistent and scalable climate reasoning annotations. This design provides preliminary supervision where large expert-annotated datasets do not exist.

- **Locational Generalizability.** Although sourced from an archive organization in Southern Quebec, the collection contains **substantial international reporting spanning multiple continents**. We provide cases to clarify that the benchmark is not region-specific.


## Paper Supplements
Most reviews focused on clarifying details or discussing further improvements. We supplement our responses in the paper:

- add a clearer explanation of the WeatherArchive-Assessment construction process in Section 3.2.2 and Section 4.1 [Reviewer vqED, CZQs].
- expand retrieval+QA experiments on raw OCR data to show performance degradation and motivate post-OCR correction [Reviewers CZQs, vWhU].
- include broader comparisons across multiple retriever–LLM combinations [Reviewers CZQs, vqED].
- provide representative global samples in Appendix L to demonstrate world-wide coverage [Reviewer P5uj].
- add a case-study analysis of how historical terminology affects retrieval performance [Reviewer vWhU].
- move the GPT-4.1 judge evaluation into the main text to highlight its importance for factual accuracy [Reviewer vqED].

We sincerely thank all the reviewers for their valuable and constructive comments, which greatly help us improve the quality of our manuscript.

Best regards,

Authors

---

### Meta-Review · Area_Chair_CoXE · 2026-01-10

**Summary:**

The paper introduces WeatherArchive-Bench, a curated corpus (with about 1M segments) and benchmark intended to evaluate retrieval-augmented reasoning over historical weather news archives, with a novel emphasis on extracting societal vulnerability and resilience signals from narrative evidence. Reviewers broadly agree the data resource and problem framing are valuable and could seed under-explored work at the intersection of RAG and climate-impact analysis.

However, multiple substantive concerns remain about benchmark validity and claim alignment. The main unresolved issues are: (1) the use of GPT-generated oracle labels, which raises circularity concerns and is exacerbated by only moderate inter-annotator agreement, making "human-level or surpassing human” interpretations inappropriate; (2) the decision to LLM-correct optical character recognition errors at scale, which some reviewers view as conflicting with the stated motivation of noisy archival text and potentially limiting real-world generalization; (iii) clarity around whether the evaluation is truly end-to-end RAG, though the rebuttal helps by detailing retrieval to QA ablations and propagation effects.

Overall, while the rebuttal addresses several presentation and experimental-setup confusions, the paper still needs a more defensible gold-standard evaluation strategy (or clearer scoping of what is and is not being benchmarked) to support its strongest claims.

**Reviewer Concerns:**

The main unresolved issues are (i) the decision to LLM-correct OCR at scale, which some reviewers view as conflicting with the stated motivation of noisy archival text and potentially limiting real-world generalization; (ii) the use of GPT-generated “oracle” labels/answers (even with human verification), which raises circularity concerns and is exacerbated by only moderate inter-annotator agreement, making “human-level/surpassing human” interpretations inappropriate; and (iii) clarity around whether the evaluation is truly end-to-end RAG.

The rebuttal helps the last point by detailing retrieval to QA ablations and propagation effects.

**Reviewer Scores:**

I don't see the reviewers changing their scores.

---

### Decision · Program_Chairs · 2026-01-26

Reject